# Exploring structural dynamics of a membrane protein by combining bioorthogonal chemistry and cysteine mutagenesis

Kanchan Gupta, Gilman ES Toombes, Kenton J Swartz*

Molecular Physiology and Biophysics Section, Porter Neuroscience Research Center, National Institute of Neurological Diseases and Stroke, National Institutes of Health, Bethesda, United States

**Abstract** The functional mechanisms of membrane proteins are extensively investigated with cysteine mutagenesis. To complement cysteine-based approaches, we engineered a membrane protein with thiol-independent crosslinkable groups using azidohomoalanine (AHA), a non-canonical methionine analogue containing an azide group that can selectively react with cycloalkynes through a strain-promoted azide-alkyne cycloaddition (SPAAC) reaction. We demonstrate that AHA can be readily incorporated into the Shaker Kv channel in place of methionine residues and modified with azide-reactive alkyne probes in *Xenopus* oocytes. Using voltage-clamp fluorometry, we show that AHA incorporation permits site-specific fluorescent labeling to track voltage-dependent conformational changes similar to cysteine-based methods. By combining AHA incorporation and cysteine mutagenesis in an orthogonal manner, we were able to site-specifically label the Shaker Kv channel with two different fluorophores simultaneously. Our results identify a facile and straightforward approach for chemical modification of membrane proteins with bioorthogonal chemistry to explore their structure-function relationships in live cells.

DOI: https://doi.org/10.7554/eLife.50776.001

*For correspondence:
swartzk@ninds.nih.gov

## Introduction

Membrane proteins play fundamental roles in maintaining cellular homeostasis by transporting ions and organic molecules and triggering intracellular signaling pathways in response to external stimuli. One of the most commonly used methods to probe structural relationships and functional dynamics of membrane proteins is the chemical modification of cysteine residues, commonly known as the substituted cysteine accessibility method (SCAM) (*Akabas, 2015*; *Akabas et al., 1992*; *Altenbach et al., 1990*; *Falke and Koshland, 1987*). The distinct reactivity of the thiol group (pKa 8.5) towards methanethiosulfonate (MTS) or maleimide conjugated reagents, low abundance in membrane proteins and the ease of introducing cysteine residues through site-directed mutagenesis, has facilitated the application of cysteine mutagenesis to study the conformation dynamics of a variety of membrane proteins including ion channels (*Akabas, 2015*; *Bezanilla, 2000*; *El Hiani and Linsdell, 2014*; *Forman and Miller, 2011*; *Horn, 2002*; *McNally and Prakriya, 2012*; *Nakajo and Kubo, 2015*; *Nys et al., 2013*), transport proteins (*Javitch, 1998*; *Mulligan and Mindell, 2017*; *Rudnick, 2006*; *Schmidt-Rose and Jentsch, 1997*; *Takeuchi et al., 2009*; *Zhu and Casey, 2007*) and G-protein coupled receptors (*Liapakis et al., 2001*; *Wess et al., 2008*).

Traditionally, cysteine mutagenesis experiments have been carried out by assessing either cysteine accessibility and/or disulfide and metal bridging in the presence or absence of a substrate or activating stimuli (*Linsdell, 2015*). Cysteine mutagenesis also provides a reactive chemical handle for

**eLife digest** Living cells can sense cues from their environment via molecules located at the interface between the inside and the outside of the cell. These molecules are mostly proteins and are made up of building blocks known as amino acids. To understand how these proteins work, fluorescent probes can be attached to amino acids within them – which can then tell when different parts of proteins move in response to a signal. Scientists often target fluorescent probes at the amino acid cysteine, because it has a chemically reactive side group and is rare enough so that unique positions can be labeled in the protein of interest. However, being able to target other amino acids would allow scientists to ask, and potentially solve, more precise questions about these proteins.

Methionine is another amino acid that has a low abundance in most proteins. Previous research has shown that the cell's normal protein-building machinery can incorporate synthetic versions of methionine into proteins. This suggested that the introduction of chemically reactive alternatives to methionine could offer a way to label membrane proteins with fluorescent probes and free up the cysteines to be targeted with other approaches. Gupta et al. set out to develop a straightforward method to achieve this and started with a well-studied membrane protein, called Shaker, and cells from female African clawed frogs, which are widely used to study membrane proteins.

Gupta et al. found that the cells could readily take up a chemically reactive methionine alternative called azidohomoalanine (AHA) from their surrounding solution and incorporate it within the Shaker protein. The AHA took the place of the methionines that are normally found in Shaker, and just like in cysteine-based methods, fluorescent probes could be easily attached to the AHAs in this membrane protein. Shaker is a protein that allows potassium ions to flow across the cell membrane by changing shape in response to the membrane voltage. The fluorescence from those probes also changed with the membrane voltage in a way that was comparable to cysteine-mediated approaches. This indicated that the AHA modification could also be used to track structural changes in the Shaker protein. Finally, Gupta et al. showed that AHA- and cysteine-mediated labeling approaches could be combined to attach two different fluorescent probes onto the Shaker protein.

This method will expand the toolbox for researchers studying the relationship between the structure and function of membrane proteins in live cells. In future, it could be applied more widely once the properties of the fluorescent probes for AHA-mediated labeling can be optimized.

DOI: https://doi.org/10.7554/eLife.50776.002

site-specifically installing fluorophores or spin labels into membrane proteins to track their conformational transitions with voltage-clamp fluorometry (*Cha and Bezanilla, 1997*; *Chanda et al., 2004*; *Gorraitz et al., 2017*; *Hou et al., 2017*; *Mannuzzu et al., 1996*; *Priest and Bezanilla, 2015*), fluorescence resonance energy transfer (FRET) (*Cha et al., 1999*; *Chanda et al., 2005*; *Glauner et al., 1999*; *Jarecki et al., 2013*; *Ji et al., 2016*; *Posson and Selvin, 2008*), electron paramagnetic resonance and double electron-electron resonance spectroscopy (*Altenbach et al., 1990*; *Paz et al., 2018*; *Pliotas, 2017*). With cysteine providing a single reactive group to investigate the conformational dynamics of membrane proteins, it becomes challenging to study multiple allosteric transitions underlying the function of the protein. Recently, a variety of non-canonical amino acids have been incorporated into membrane proteins to introduce reactive chemical groups, environmentally sensitive biophysical reporters and subtle chemical modifications (*Huber and Sakmar, 2014*; *Klippenstein et al., 2018*; *Pless and Ahern, 2013*; *Rannversson et al., 2016*; *Van Arnam and Dougherty, 2014*; *Young and Schultz, 2018*). Typically, non-canonical amino acids are incorporated through a nonsense suppression method using chemically charged tRNA (*Nowak et al., 1995*) or a specific pair of tRNA and amino acyl-tRNA synthetase (*Noren et al., 1989*). Since non-canonical amino acid mutagenesis is compatible with cysteine mutagenesis, the two have been combined to expand the scope and precision of strategies for studying membrane proteins (*Dai et al., 2019*; *Gordon et al., 2018*; *Kalstrup and Blunck, 2013*). However, the success rate and efficiency of non-canonical amino acid mutagenesis varies considerably with the type of membrane protein being investigated, expression system, choice of the non-canonical amino acid and the target site in the protein (*Kalstrup and Blunck, 2015*; *Leisle et al., 2015*; *Pless et al., 2015*).

The goal of the present study was to develop a facile and generalizable strategy to incorporate a bioorthogonal (a chemical group which is absent in vivo) crosslinkable amino acid into membrane proteins for installation of biophysical reporters that could then be combined with cysteine mutagenesis for the arsenal of approaches outlined above. We used an alternate to the nonsense suppression approach where an essential amino acid is simply replaced by supplying excess of an non-canonical analogue, resulting in the global replacement of the natural amino acid in the newly synthesized protein (*Budisa, 2004*; *Johnson et al., 2010*; *Link and Tirrell, 2005*). One of the most successful examples of this strategy is the replacement of methionine residues by incorporation of selenomethionine into proteins for X-ray crystallography (*Cohen and Cowie, 1957*; *Saotome et al., 2016*; *Yang et al., 1990*). Incorporation of methionine analogues into proteins has been particularly successful due to the unique conformational flexibility in the amino acid binding site of methionyl tRNA synthetase (*Nadarajan et al., 2013*; *van Hest and Tirrell, 1998*) and indispensable dependence of most eukaryotic systems on external sources for this essential amino acid. To install a bioorthogonally crosslinkable amino acid, we chose azidohomoalanine (AHA) (*Figure 1A*) (*Kiick et al., 2002*), a non-canonical amino acid that is nearly isosteric with methionine (*Figure 1A*) and contains an azide group capable of reacting selectively with strained alkynes (*Figure 1B*). Indeed, AHA has previously been incorporated into proteins to identify and visualize newly synthesized proteins after attachment of biotin (*Dieterich et al., 2006*) or fluorescent probes (*Dieterich et al., 2010*) using bioorthogonal azide-alkyne cycloaddition reactions (*Agard et al., 2004*; *Rostovtsev et al., 2002*; *Tornøe et al., 2002*), approaches that have been successful in both prokaryotic and eukaryotic systems (*Ma and Yates, 2018*).

Using the Shaker voltage-activated potassium (Kv) channel as a test case, we find that AHA can be readily incorporated into the channel and enables site-specific installation of fluorophores in the protein via a catalyst-independent strain-promoted azide-alkyne cycloaddition (SPAAC) reaction in live *Xenopus laevis* oocytes (*Figure 1B*) (*Agard et al., 2004*). We show that fluorophores attached to AHA can be used to track conformational changes of the voltage-sensor in a manner analogous to cysteine-based methods. Finally, we implement a straightforward strategy to carry out two-color labeling of membrane proteins in a site-specific manner using a combination of AHA incorporation and cysteine mutagenesis.

## Results

### Incorporation of AHA into the Shaker Kv channel

We began by testing whether AHA could be introduced into the Shaker Kv channel, an extensively studied ion channel protein that opens and closes in response to changes in membrane voltage (*Bezanilla, 2000*; *Horne and Fedida, 2009*; *Swartz, 2008*). Shaker is an oligomeric integral membrane protein containing four identical subunits, with each subunit containing six transmembrane (TM) segments. The S1-S4 segments from each subunit form peripheral voltage-sensing domains, while the S5-S6 segments from all four subunits constitute the central pore domain (*Long et al., 2007*) (*Figure 1C*). Due to a close homology with other eukaryotic Kv channels and optimal expression in heterologous expression systems, Shaker has been extensively subjected to a variety of chemical modifications through cysteine mutagenesis (*Gandhi et al., 2003*; *Gonzalez et al., 2005*; *Gross and MacKinnon, 1996*; *Holmgren et al., 1998*; *Horne and Fedida, 2009*; *Larsson et al., 1996*; *Liu et al., 1996*). In addition, many non-canonical amino acids have been successfully incorporated into the Shaker Kv channel (*Infield et al., 2018*; *Kalstrup and Blunck, 2013*; *Pless et al., 2015*; *Tao et al., 2010*).

We initially examined whether AHA could be incorporated in the Shaker Kv channel expressed in *Xenopus* oocytes using surface biotinylation to detect AHA incorporation into the channel (*Figure 1—figure supplement 1*). The Shaker Kv channel contains 12 methionine residues in each subunit, 5 of which are in TM regions of the channel (*Figure 1C*, cyan) (*Long et al., 2007*). Cysteine accessibility experiments have shown that two out of five methionine residues in the TM regions of Shaker (M356 and M448) face the extracellular side of the channel (*Figure 1C*) (*Larsson et al., 1996*; *Liu et al., 1996*), providing the means of detecting AHA incorporation into the Shaker Kv channel expressed on the surface of live *Xenopus* oocytes. In eukaryotic expression systems, AHA is typically incorporated into proteins by adding the non-canonical amino acid into methionine-free cell culture

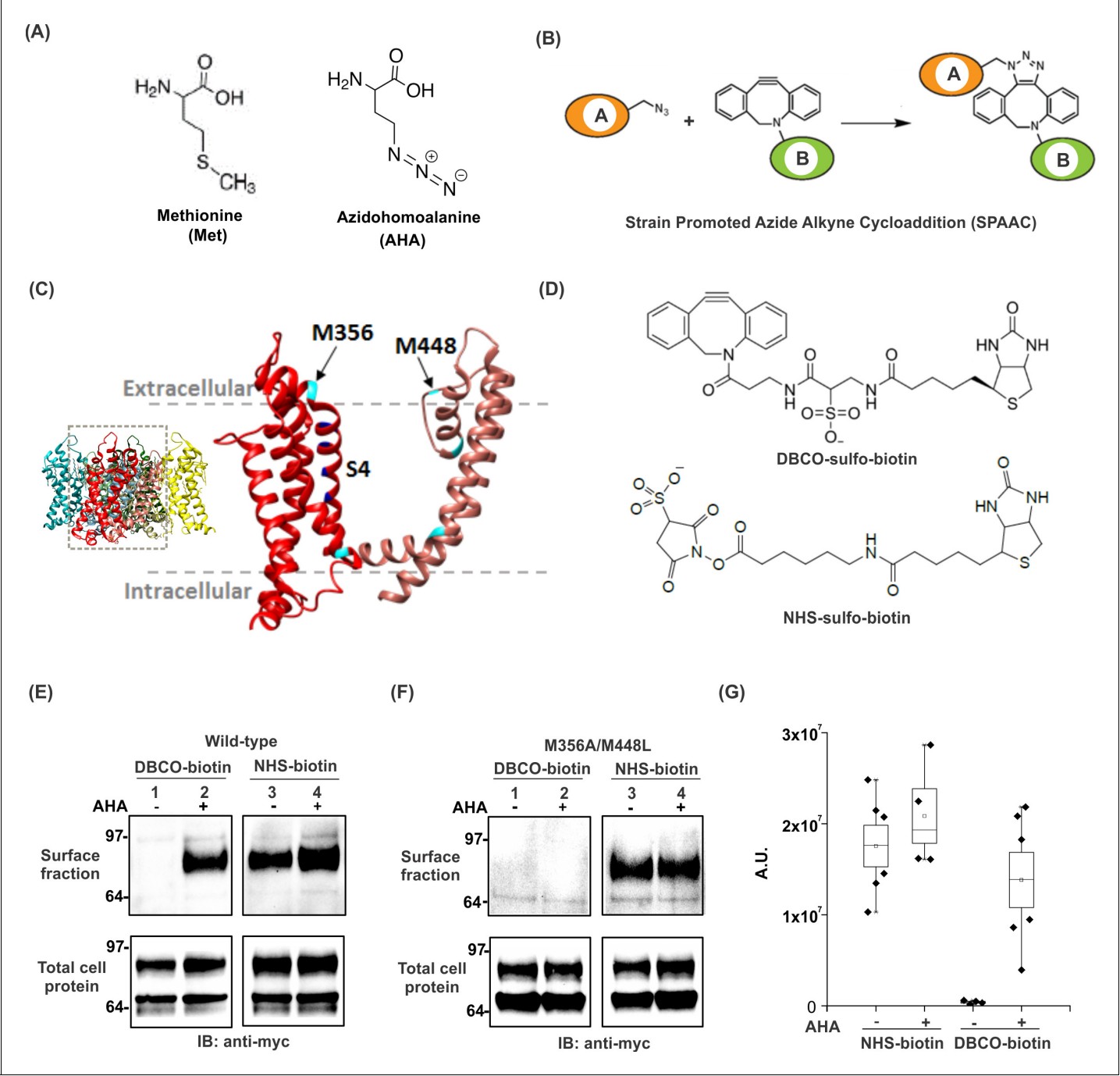

**Figure 1.** Incorporation and detection of azidohomoalanine into the Shaker Kv channel. (**A**) Structures of Methionine and Azidohomoalanine. (**B**) A schematic for the strain promoted azide alkyne cycloaddition (SPAAC) reaction. (**C**) Transmembrane region of a single subunit of a Kv channel containing voltage sensing domain (red) and pore domain (pink). Methionine residues are colored in cyan. Inset shows tetrameric structure of the Kv1.2–2.1 paddle chimera crystal structure, 2R9R (*Long et al., 2007*). (**D**) Structures of biotin probes; DBCO-sulfo-biotin (top) and NHS-sulfo-biotin (bottom). (**E–F**) Anti-myc western blots for the surface fraction (top) and total cell protein (bottom) isolated from *Xenopus laevis* oocytes injected with ShakerΔ5-V478W-myc (Wild-type, **E**) or the mutant lacking the methionine residues facing the extracellular side (M356A/M448L, **F**). (**G**) Densitometry plots of anti-myc western blots for the surface fraction of the wild-type Shaker Kv channel in the absence or presence of AHA. A.U. refers to arbitrary units for absolute chemiluminescence intensity. Boxes represent SEM for n = 4–6. The small open squares and black horizontal lines represent the mean and weighted mean values, respectively, for the chemiluminescence intensity. Vertical black lines represent the full range of data.

DOI: https://doi.org/10.7554/eLife.50776.003

The following figure supplements are available for figure 1:

*Figure 1 continued on next page*

*Figure 1 continued*

**Figure supplement 1.** Scheme for detecting AHA incorporation into the Shaker Kv channel.
DOI: https://doi.org/10.7554/eLife.50776.004
**Figure supplement 2.** Densitometry plots of anti-myc western blots for the total cell protein obtained after cell lysis from oocytes injected with the Shaker∆5-V478W-myc (wild-type, *Figure 1E*) in the absence and presence of AHA.
DOI: https://doi.org/10.7554/eLife.50776.005

medium at concentrations ranging from 1 to 4 mM (*Dieterich et al., 2006*), but has not been previously tested with *Xenopus* oocytes, a commonly used expression system for studying membrane proteins. To incorporate AHA into the Shaker Kv channel, we preincubated oocytes overnight with 4 mM AHA in the ND96 maintenance buffer to compete out the endogenous pool of free methionine. The next day, cRNA for a myc-tagged construct of the Shaker Kv channel was injected (See Materials and methods), and oocytes were maintained in the presence of AHA. After 3–4 days at 17˚C, excess AHA was removed by washing oocytes with ND96 and then incubated with membrane-impermeable biotinylation reagents, either azide-reactive dibenzocyclooctyne (DBCO)-sulfo-biotin (*Figure 1D*, top) to tag AHA-modified proteins or amine-reactive NHS-sulfo-biotin (*Figure 1D*, bottom) to tag all proteins expressed on the cell surface. After removing unreacted biotin probes by washing with ND96, oocytes were lysed with a triton X-100 containing lysis buffer, followed by neutravidin agarose pull-down. Subsequently, we analyzed both the cell lysate (total cell protein) and the surface fraction with anti-myc western blotting. AHA-supplemented oocytes showed a single band for the mature Shaker Kv channel subunits in the surface fraction isolated with DBCO-sulfo-biotin (*Figure 1E*, surface fraction, lane 2), indicating successful incorporation of AHA into the Shaker Kv channel. No pull-down was observed with DBCO-sulfo-biotin in the absence of AHA (*Figure 1E*, surface fraction, lane 1), although the protein expression was similar in both the cases (*Figure 1E*, total cell protein, lane 1 and 2), demonstrating that the DBCO is chemically selective and does not react with other residues of the protein. In contrast, pull down with NHS-sulfo-biotin yielded similar amounts of protein both in the absence and presence of AHA (*Figure 1E*, surface fraction and total cell protein, lane 3 and 4), indicating that introduction of AHA into the Shaker Kv channel has no detectable effect on expression and trafficking of the channel in *Xenopus* oocytes. Moreover, AHA supplementation had no toxic effects on the survival of oocytes in ND96 at 17˚C (data not shown), consistent with previous reports using this non-canonical amino acid in other expression systems (*Dieterich et al., 2006*; *Hinz et al., 2013*).

Next, we tested whether AHA replaces only methionine residues in the Shaker Kv channel when expressed in *Xenopus* oocytes by mutating the two methionine residues on the extracellular side of the Shaker Kv channel to Ala (M356A) or Leu (M448L) (*Figure 1C*) (*Larsson et al., 1996*; *Liu et al., 1996*). When oocytes expressing the M356A/M448L mutant channel were labeled with DBCO-sulfo-biotin, no detectable pull-down was observed in the absence or presence of AHA (*Figure 1F*, surface fraction, lane 1 and 2), although a comparable amount of protein expression was observed in the cell lysate (*Figure 1F*, total cell protein, lane 1 and 2), indicating that the expression of mutant Shaker Kv channel was not affected substantially. In contrast, NHS-sulfo-biotin showed a similar amount of pull down for the mutant protein in both cases (*Figure 1F*, surface fraction and total cell protein, lane 3 and 4), establishing that AHA specifically replaces methionine residues in membrane proteins expressed in *Xenopus* oocytes. These results unambiguously demonstrate that *Xenopus* oocytes can uptake AHA from the extracellular medium and their endogenous protein synthesis machinery can robustly incorporate this non-canonical amino acid into newly expressed heterologous proteins.

To assess the efficiency of AHA incorporation into the Shaker Kv channel, we used densitometry to analyze the anti-myc western blots from wild-type protein (*Figure 1E*). In general, AHA incorporation had no detectable effect on the total protein expression of the Shaker Kv channel (*Figure 1— figure supplement 2*). Moreover, AHA supplementation does not alter the surface expression of the Shaker Kv channel, as the amount of protein pulled down with NHS-sulfo-biotin, which provides an estimate for the total expression of Shaker on the surface of oocytes, was similar in the absence and presence of AHA (*Figure 1G*). In contrast, DBCO-sulfo-biotin pulls down a comparable amount of protein in the presence of AHA and fails to pull down any protein in the absence of AHA in repeated trials (*Figure 1G*). We did observe some variability in the pull down with DBCO-sulfo-biotin as

compared to NHS-sulfo-biotin, suggesting that the extent of AHA incorporation can vary between different batches of oocytes.

## AHA incorporation does not alter the gating properties of the Shaker Kv channel

AHA-modification of the Shaker Kv channel will replace up to 12 methionine residues with the non-canonical amino acid in regions of the protein known to be critical for voltage-dependent gating (*Figure 1C*) (*Bezanilla, 2008*; *Swartz, 2008*). To explore whether AHA incorporation alters the gating behavior of the Shaker Kv channel, we initially expressed the channel in the absence and presence of AHA and used the two-electrode voltage clamp recording technique to obtain voltage-activation relationships from macroscopic ionic currents with $K^+$ as the charge carrier (See Materials and methods). When the membrane voltage was stepped between −100 mV and +50 mV, we observed robust voltage-activated $K^+$ currents in the presence and absence of AHA, with voltage-activation relationships that were not discernably different (*Figure 2A,B*), suggesting that AHA incorporation does not detectably alter the overall process of voltage-dependent gating in the Shaker Kv channel. Activation of the voltage-sensing domains in the Shaker Kv channel involves the movement

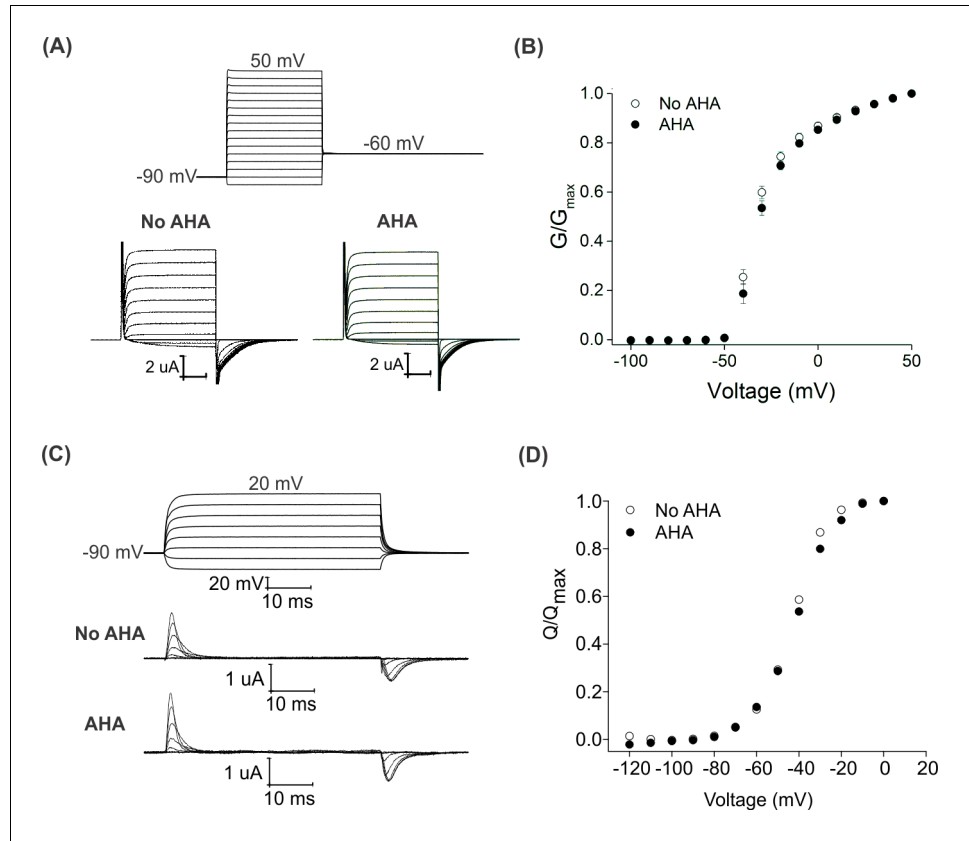

**Figure 2.** Effect of AHA on the gating behavior of the Shaker Kv channel. (**A**) Ionic currents elicited by voltage steps for oocytes injected with Shaker-IR in the absence (left) or presence of AHA (right). Holding voltage = −90 mV, tail voltage = −60 mV. (**B**) G-V relationships obtained from tail currents at −60 mV in the absence (open circles) and presence of AHA (closed circles). All data points represent mean ± SEM (n = 3) (**C**) Gating currents obtained from Shaker-V478W, a non-conducting mutant of Shaker, in the absence (top) or presence of AHA (bottom). (**D**) Q-V relationships obtained from the gating currents elicited after stepping to different voltages from a holding voltage of −90 mV in the absence (open circles) and presence of AHA (closed circles).
DOI: https://doi.org/10.7554/eLife.50776.006

The following figure supplement is available for figure 2:

**Figure supplement 1.** Effect of AHA on the gating behavior of the Shaker Kv channel.
DOI: https://doi.org/10.7554/eLife.50776.007

of positively charged arginine residues, which can be directly measured as non-linear capacitive currents known as gating currents (*Bezanilla et al., 1991*). Using the V478W non-conducting mutant of the Shaker Kv channel (*Hackos et al., 2002*; *Kitaguchi et al., 2004*), we also measured gating currents in the presence and absence of AHA and obtained gating charge (Q)-voltage (V) relationships that were similar (*Figure 2C,D*), suggesting that voltage-sensor activation was also not detectably altered with AHA incorporation. Finally, we tested whether AHA incorporation alters the sensitivity of the Shaker Kv channel to a tarantula toxin that binds to the S1-S4 voltage-sensing domain to allosterically inhibit opening of the channel in response to membrane depolarization. Using a construct of the Shaker Kv channel that is sensitive to the tarantula toxin GxTx1E (ShakerΔ5) (*Herrington et al., 2006*; *Milescu et al., 2009*; *Milescu et al., 2013*), we observed that the toxin produced robust and indistinguishable shifts of voltage-activation relationships to more positive voltages, both in the absence and presence of AHA (*Figure 2—figure supplement 1*). Collectively, these results demonstrate that global replacement of methionine residues with AHA in the Shaker Kv channel does not result in detectable alterations in the gating properties of the channel.

## Site-specific installation of fluorophores into AHA-modified Shaker Kv channels

After successfully labeling the Shaker Kv channel with cyclooctyne-conjugated biotin probes through AHA, we explored the possibility of using the SPAAC reaction to install environmentally-sensitive fluorophores in a site-specific manner and monitor the conformational dynamics of the channel using voltage-clamp fluorometry (VCF) (*Cha and Bezanilla, 1997*; *Mannuzzu et al., 1996*). This technique has been widely used to investigate a variety of membrane proteins, including the Shaker Kv channel (*Horne and Fedida, 2009*), after installation of different thiol-reactive fluorophores using cysteine mutagenesis (*Priest and Bezanilla, 2015*). For site-specific fluorescent labeling, we generated the Shaker-M356 construct (M356/M448L), where the methionine residue in the pore domain (M448) was mutated to leucine, leaving a single accessible methionine residue (M356) on the extracellular side of the channel. M356 is located at the N-terminus of the S4 helix within the voltage-sensing domain (*Figure 1C*), is accessible to extracellular thiol-reactive compounds when mutated to cysteine (*Cha and Bezanilla, 1997*; *Larsson et al., 1996*; *Mannuzzu et al., 1996*) and fluorophores attached at this position exhibit changes in fluorescence as the protein undergoes conformational changes in response to changes in membrane voltage (*Cha and Bezanilla, 1997*; *Mannuzzu et al., 1996*). To facilitate gating current measurements, all voltage clamp fluorometry experiments were carried out with the V478W non-conducting mutant. Since the advent of the SPAAC reaction, a variety of cyclooctynes with varying size, hydrophobicity and reactivity towards the azide group have been synthesized (*Dommerholt et al., 2016*; *Sletten and Bertozzi, 2011*). In addition, many cyclooctyne conjugated fluorophores have been generated and are commercially available (*Supplementary files 1* and *2*), although in some instances their solubility is limited in aqueous solutions (e.g. TAMRA-DBCO). In order to maximize the aqueous solubility of cyclooctyne-fluorophore conjugate, we chose to work with a relatively polar and charged fluorophore, Alexa 488 (*Hughes et al., 2014*).

We began by measuring baseline fluorescence signals at a holding voltage of −90 mV using a filter cube appropriate for Alexa 488 (ex. 480/40 nm; em. 535/50 nm) after labeling with a cyclooctyne-conjugated Alexa 488 fluorophore (AF488-DBCO; *Figure 3A*) and compared uninjected oocytes with those expressing Shaker-M356 in the absence and presence of AHA (*Figure 3B*). In the absence of AHA, we observed a 2-fold increase in fluorescence intensity with both uninjected and Shaker-M356 expressing oocytes when compared to unlabeled oocytes (*Figure 3B*, Uninjected and M356), which may reflect non-specific interactions between the oocyte membrane and the hydrophobic cyclooctyne group in AF488-DBCO. In the presence of AHA, labeling of uninjected oocytes showed a 5-fold increase in the fluorescence intensity, suggesting incorporation of AHA into a fraction of endogenous oocyte proteins during basal protein turnover (*Figure 3B*, Uninjected + AHA). In contrast, oocytes expressing AHA-modified Shaker-M356 exhibited ~15 fold higher baseline fluorescence intensity compared to unlabeled oocytes (*Figure 3B*, Shaker-M356 + AHA), indicating that AHA modification enables the preferential labeling of newly synthesized Shaker Kv channels with cyclooctyne-conjugated fluorophores over other endogenous proteins in *Xenopus* oocytes. In addition, gating currents measured from oocytes with and without labeling with AF488-DBCO reveals that the Q-V relationship of labeled oocytes was detectably shifted towards more depolarized

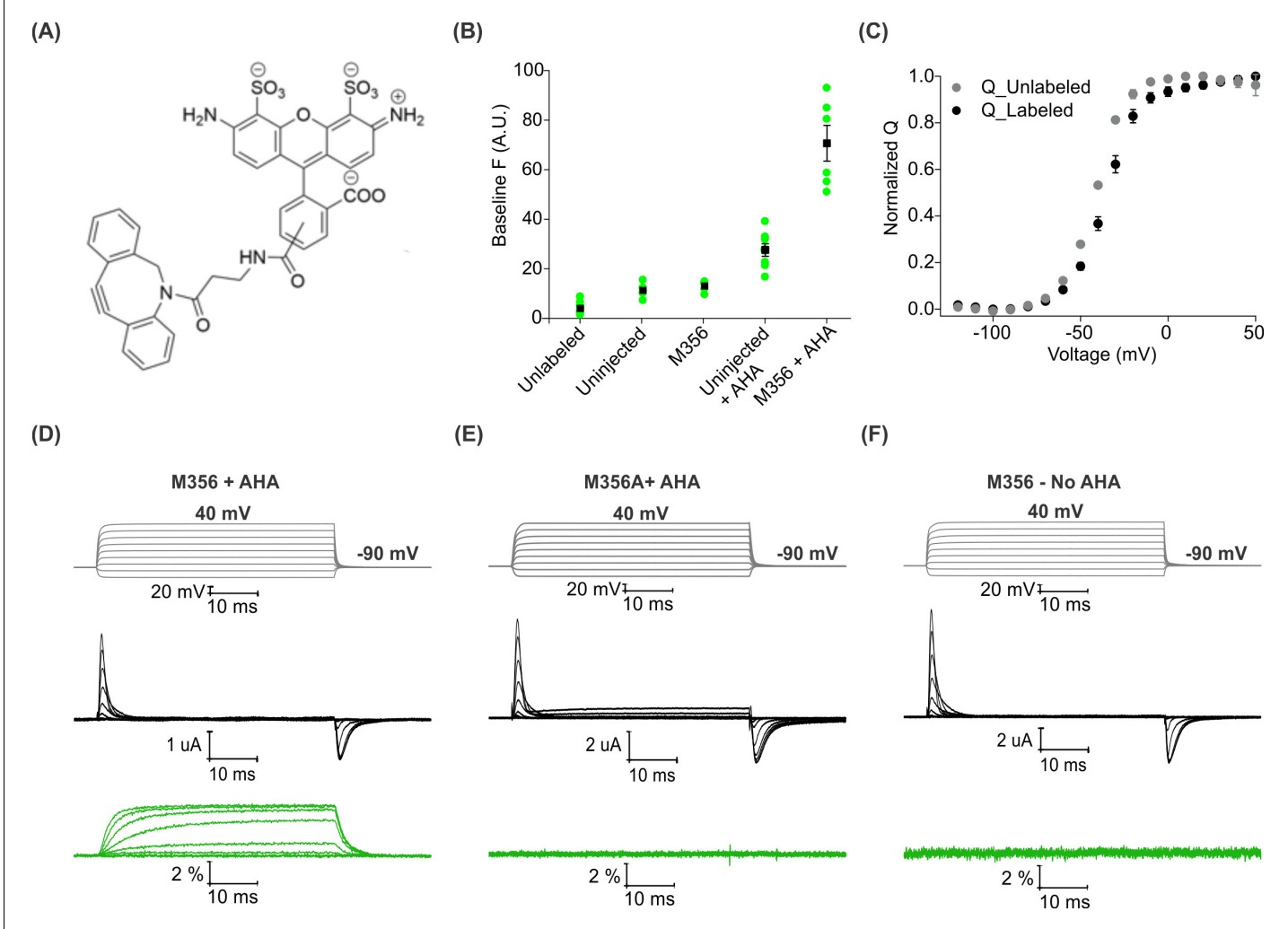

**Figure 3.** Voltage clamp fluorometry with AHA-modified Shaker Kv channels. (A) Structure of azide reactive cyclooctyne-conjugated Alexa Fluor 488, AF488-DBCO. (B) Baseline fluorescence intensity (at −90 mV) obtained from unlabeled and labeled oocytes, either uninjected or injected with Shaker-M356 (M356/M448L) in the presence or absence of AHA (n = 5–6) (C) Q-V relationship obtained from AHA-modified Shaker-M356 before (gray) and after (black) labeling with AF488-DBCO. All data points represent mean ± SEM (n = 3). (D–F) Representative signals for gating currents (black) and fluorescence responses (green) obtained from AF488-DBCO labeled oocytes, injected with Shaker-M356 (D) or M356A (Shaker-M356A/M448L) (E) in the presence of AHA or Shaker-M356 in the absence of AHA (F).
DOI: https://doi.org/10.7554/eLife.50776.008

voltages (*Figure 3C*), which is consistent with a bulky and charged fluorophore attaching to and modulating the movement of S4 helix (*Cha and Bezanilla, 1997*).

Next, we investigated whether voltage-sensor activation would produce a change in the fluorescence of AF488-DBCO labeled oocytes expressing Shaker-M356. As the oocyte membrane was depolarized to positive voltages from a holding voltage of −90 mV, we observed readily detectable increase in fluorescence signals that saturated at positive membrane voltages and returned to the baseline value on membrane repolarization (*Figure 3D*). We observed no detectable change in fluorescence with AF488-DBCO labeled oocytes expressing the M356A mutant (M356A/M448L) in the presence of AHA (*Figure 3E*), demonstrating that the voltage-dependent fluorescence response specifically originates from AF488-DBCO conjugated at the M356 position in the Shaker Kv channel. Similarly, oocytes expressing Shaker-M356 in the absence of AHA and labeled with AF488-DBCO did not produce any change in fluorescence on membrane depolarization (*Figure 3F*). Taken together, these results demonstrate that AHA-modified Shaker Kv channels can be labeled with

azide-reactive fluorophores in a site-specific and chemoselective manner to track the conformational changes of the channel in response to changes in membrane voltage.

## Comparing fluorescence responses with fluorophores installed into Shaker using AHA and cysteine

Having established AHA as a unique chemical handle for site-specific fluorescent labeling of the Shaker Kv channel, we wanted to compare the properties of AHA-mediated fluorescent labeling of the channel with the well-established cysteine-based method. To monitor the fluorescence response from a single methionine or cysteine residue placed at identical sites in the protein, we designed a construct, Shaker-M356* (M356/M448L/C245V/C462A) that has two endogenous cysteines (C245V and C462A) along with the methionine (M448L) residue in the pore domain mutated. For AHA-mediated labeling, we used the native methionine at M356 position and for the cysteine-mediated labeling, we mutated this methionine to a cysteine and generated the Shaker-M356C construct (M356C/ M448L/C245V/C462A). We chose AF488-C5-maleimide to carry out cysteine mediated labeling as the linker length in the two probes is similar (*Figure 4—figure supplement 1*). Although our comparison was done using the same fluorophore, labeling sites and background constructs, the residue at the labeling site and attachment chemistries are necessarily different, and therefore the results are not expected to be identical. In addition, the M356C mutant of the Shaker Kv channel has been shown to slow voltage sensor activation and shift the Q-V relationship to more positive voltages (*Cha and Bezanilla, 1997*), perturbations not observed with replacement of M356 with AHA (*Figure 2C,D*). Oocytes injected with Shaker-M356* and Shaker-M356C were labeled with the complementary azide or thiol-reactive fluorophores using identical protocols (See Materials and methods). For both AF488-DBCO and AF488-C5-maleimide labeled oocytes, we observed an increase in fluorescence intensity with membrane depolarization that saturated at positive membrane voltages (*Figure 4A,B*), indicating that the labeling chemistry does not affect the qualitative behavior of the fluorophore in response to voltage-dependent conformational changes in the Shaker Kv channel.

We characterized the behavior of the two fluorescent probes with respect to the gating behavior of the Shaker Kv channel and analyzed the relationship between gating charge movement and changes in fluorescence intensity in each case. The steady state F-V relationship obtained from the labeled Shaker-M356* construct exhibited a detectable shift towards depolarized voltages in comparison to the Q-V relationship (*Figure 4C*), whereas a closer overlap was observed between the Q-V and F-V relationships obtained from the labeled Shaker-M356C construct (*Figure 4D*). In addition, the onset of the fluorescence response from Shaker-M356* was discernibly slower than the displacement of gating charge upon depolarization (*Figure 4E*) but overlapped closely during repolarization (*Figure 4G*). In the case of the Shaker-M356C channel, both the gating currents and fluorescence response showed multiple kinetic components during activation and deactivation of the channel (*Figure 4F,H*). Our observations on the behavior of the AF488-C5-maleimide labeled Shaker-M356C channel are similar to those reported for the M356C mutant of Shaker after labeling with Oregon green maleimide, a thiol-reactive fluorophore with identical excitation and emission spectra to AF488 (*Cha and Bezanilla, 1997*). Taken together, this comparison shows that fluorescent labeling of AHA-modified Shaker Kv channels with cyclooctyne-conjugated fluorophores can be utilized to track the conformational rearrangements similar to cysteine-based methods.

To compare the magnitude of fluorescence responses as a function of protein expression level for fluorophore installation using AHA- and cysteine-based approaches, we measured maximal fluorescence responses ($\Delta F/F$, %) along with $Q_{max}$ to estimate the total number of Shaker Kv channels expressed on the surface of oocytes (*Aggarwal and MacKinnon, 1996*). For both AF488-DBCO and AF488-C5-maleimide labeled oocytes (*Figure 5A,B*), the magnitude of maximal fluorescence response increases along with channel expression on the surface of oocytes (*Figure 5C,D*), although there is a considerable spread in both relationships. Variability in fluorescence responses is to be expected given the heterogeneity in the endogenous oocyte fluorescence around 480 nm excitation (*Lee and Bezanilla, 2019*). Nevertheless, comparison of the trends for AHA- and cysteine-mediated fluorescent labeling suggests that AHA-mediated labeling requires approximately two-fold higher protein expression when compared to cysteine-mediated labeling (*Figure 5C,D*). This difference could arise from either incomplete incorporation of AHA and/or fluorophore labeling due to the slower rate of SPAAC reaction compared to the reaction between maleimide and cysteine (*Dommerholt et al., 2016*; *Lang and Chin, 2014*; *Saito et al., 2015*).

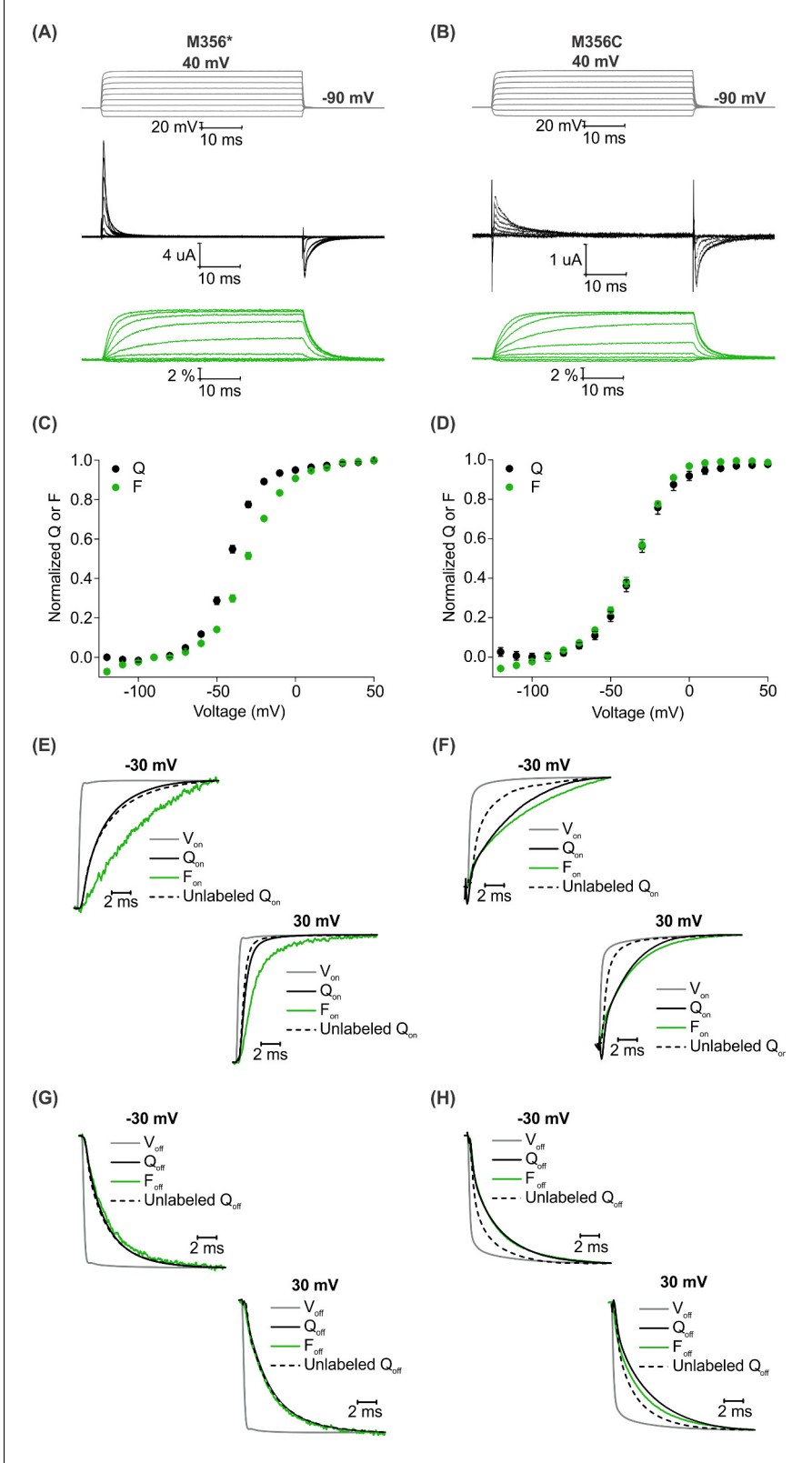

**Figure 4.** A comparison of AHA and cysteine-mediated voltage clamp fluorometry with the Shaker Kv channel. (A–B) Representative traces for gating currents (black) and fluorescence responses (green) obtained from oocytes injected with Shaker-M356* (M356/M448L/C245V/C462A) (A) or Shaker-M356C (M356C/M448L/C245V/C462A) (B) after labeling with AF488-DBCO or AF488-C5-Maleimide, respectively. (C–D) Relationship between total gating charge displaced (Q, black) and change in fluorescence intensity (F, green) at steady state as a function of voltage for oocytes injected with

*Figure 4 continued on next page*

Figure 4 continued

Shaker-M356*, n = 18 (C) or Shaker-M356C, n = 5 (D). All data points represent mean ± SEM. (E–H) Kinetics of displacement of gating charge (black) and change in fluorescence intensity (green) during activation (E–F) and deactivation (G–H) of voltage sensors at weak (−30 mV, top) and strong depolarization (30 mV, bottom). Dashed lines represent the displacement of gating charge in unlabeled oocytes. Gray traces represent the integrated capacitive transient as a measure of the speed of the voltage clamp.

DOI: https://doi.org/10.7554/eLife.50776.009

The following figure supplement is available for figure 4:

**Figure supplement 1.** Structure of thiol reactive Alexa Fluor 488, AF488-C5-Maleimide.

DOI: https://doi.org/10.7554/eLife.50776.010

## Two-color labeling of the Shaker Kv channel using a combination of AHA and cysteine

Membrane proteins are also labeled simultaneously with two distinct biophysical reporters to investigate their conformational transitions through FRET (*Taraska, 2012*; *Taraska and Zagotta, 2010*) or to independently track structural rearrangements in two different regions of the protein (*Kalstrup and Blunck, 2013*; *Kalstrup and Blunck, 2018*). Thus far, two-color labeling of membrane proteins has been achieved using pairs of cysteine residues (*Glauner et al., 1999*; *Koch, 2005*; *Posson and Selvin, 2008*; *Wang et al., 2018*), where it is difficult to monitor the site-specific attachment of fluorophores and often suffers from complexities arising from mixed populations of proteins containing one or both fluorophores. Cysteine mutagenesis has also been combined with fluorescently-labeled ligands (*Posson and Selvin, 2008*), transition metal binding sites (*Billesbølle et al., 2016*; *Taraska et al., 2009*), lanthanide metal binding peptide tags (*Vázquez-Ibar et al., 2002*) or fluorescent non-canonical amino acids (*Gordon et al., 2018*; *Kalstrup and Blunck, 2013*) to achieve site-specific labeling of membrane proteins with two different biophysical reporters. Interestingly, biorthogonal reactions including the SPAAC reaction and the copper mediated azide alkyne cycloaddition (CuAAC) reaction have also been combined with thiol-mediated reactions for two-color labeling, but exclusively with relatively small and soluble purified proteins containing azide- or alkyne-terminated amino acids introduced through the nonsense suppression method (*Sadoine et al., 2017*; *Seo et al., 2011*; *Tyagi and Lemke, 2013*). Given the straightforward nature of AHA incorporation, the biocompatible nature of the SPAAC reaction and the comparable fluorescence responses observed using AHA- and cysteine-based approaches (*Figures 4* and *5*), we explored the possibility of combining the two methods for two-color labeling of the Shaker Kv channel using azide and thiol-mediated chemical reactions in live cells.

To install two different fluorophores simultaneously into the Shaker Kv channel, we added a cysteine mutation at position S424C in the outer mouth of pore domain of Shaker-M356* to generate Shaker-M356*-S424C (M356/M448L/C245V/C462A/S424C). The S424C site is accessible to fluorescent labeling with thiol-reactive TAMRA-maleimide fluorophore (*Gandhi et al., 2000*; *Loots and Isacoff, 1998*; *Loots and Isacoff, 2000*), the resulting voltage-dependent fluorescence responses are distinct from those measured when TAMRA fluorophores are attached to the external end of S4 (*Cha and Bezanilla, 1997*) and have been proposed to report on conformational rearrangements associated with slow inactivation of the channel (*Claydon et al., 2007*; *Loots and Isacoff, 1998*). Thus, fluorophores attached at M356 in the voltage-sensing domain and S424 in the pore domain of the Shaker-M356*-S424C construct should report on distinct conformational changes in these two regions of the protein. We first measured fluorescence responses of Shaker-M356*-S424C when labeled independently with AF488-DBCO or TAMRA-MTS. Oocytes injected with Shaker-M356*-S424C in the presence of AHA gave rise to functional channels after labeling with AF488-DBCO (*Figure 6A*) or TAMRA-MTS (*Figure 6B*). AF488-DBCO labeled oocytes produced a similar fluorescence response as observed with Shaker-M356* through the 488 filter cube (ex. 480/40 nm; em. 535/50 nm) (*Figure 4A,C*), indicating that the additional cysteine mutation in the pore did not substantially affect the fluorescence behavior of AF488 installed at M356 (*Figure 6C,G*). In contrast, TAMRA-MTS labeled oocytes generated distinct fluorescence responses through the TAMRA filter cube (ex. 535/50 nm; em. 610/75 nm) when compared to the fluorophore on top of S4 and were consistent with responses reported when labeling with TAMRA-maleimide (*Figure 6F*) (*Claydon et al., 2007*). The fluorescence-voltage relationships for AF488-DBCO on S4 and TAMRA-

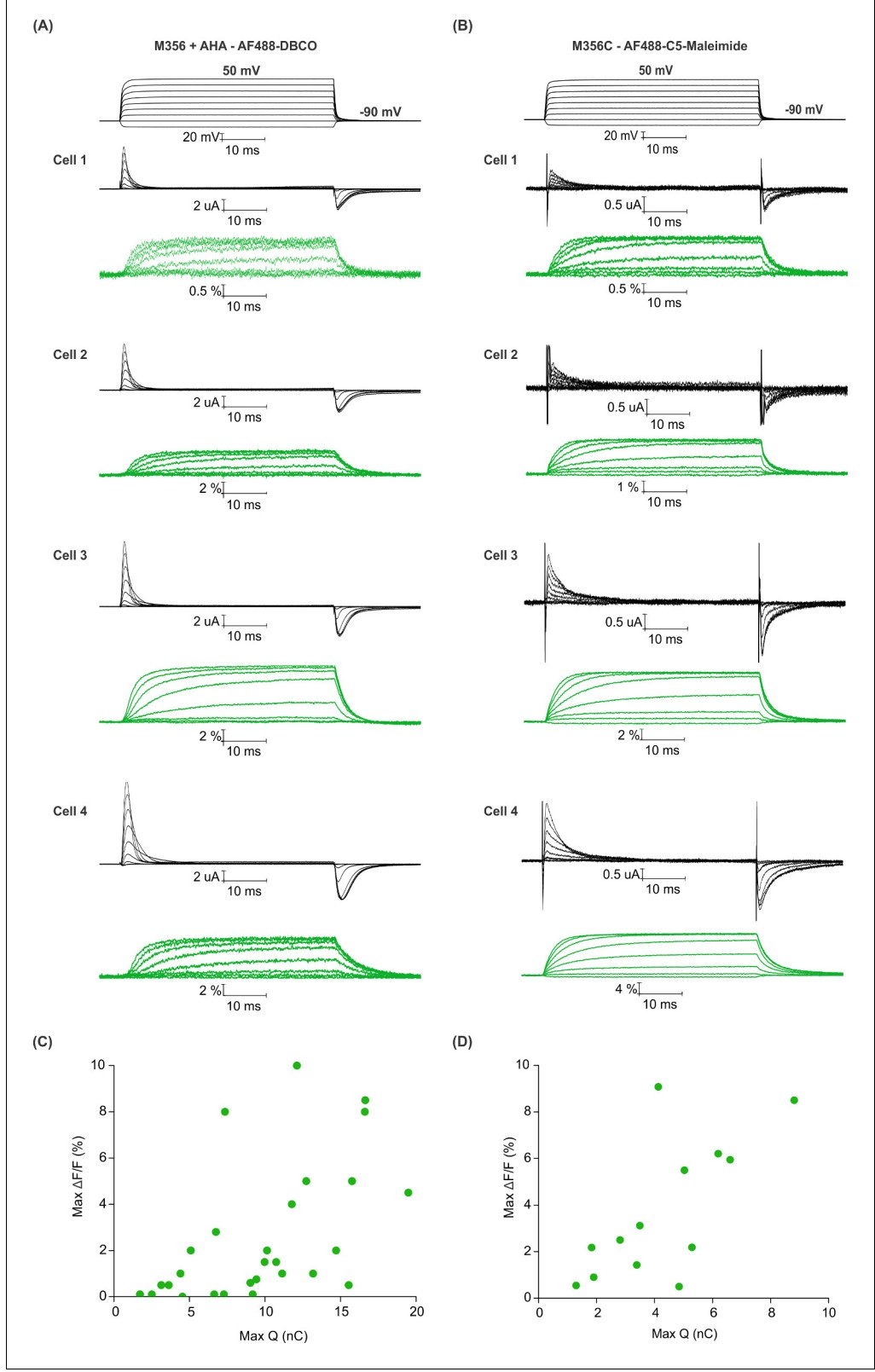

**Figure 5.** Efficiency of AHA and cysteine-mediated voltage clamp fluorometry with Shaker. (A–B) Gating currents (black) and fluorescence responses (green) from AF488-DBCO labeled Shaker-M356 (M356/M448L) in the presence of AHA (A) or AF488-C5-maleimide labeled Shaker-M356C (M356C/M448L/C245V/C462A) (B). (C–D) Scatter plot for maximum fluorescence signal (Max ΔF/F, %) obtained as a function of total gating charge displaced (Max Q, nC) for oocytes labeled at M356 through AHA (C) or cysteine (D).

*Figure 5 continued on next page*

*Figure 5 continued*

DOI: https://doi.org/10.7554/eLife.50776.011

MTS within the pore domain were radically different from each other (*Figure 6G,H*), and no fluorescence response was detected when AF488-DBCO labeled oocytes were subjected to TAMRA excitation/emission (ex. 535/50 nm; em. 610/75 nm) (*Figure 6E*) or when TAMRA-MTS labeled oocytes were subjected to AF488 excitation/emission (ex. 480/40 nm; em. 535/50 nm) (*Figure 6D*). Thus, we could clearly distinguish between the fluorescence responses originating from AF488-DBCO or TAMRA-MTS labeled Shaker-M356*-S424C in the presence of AHA.

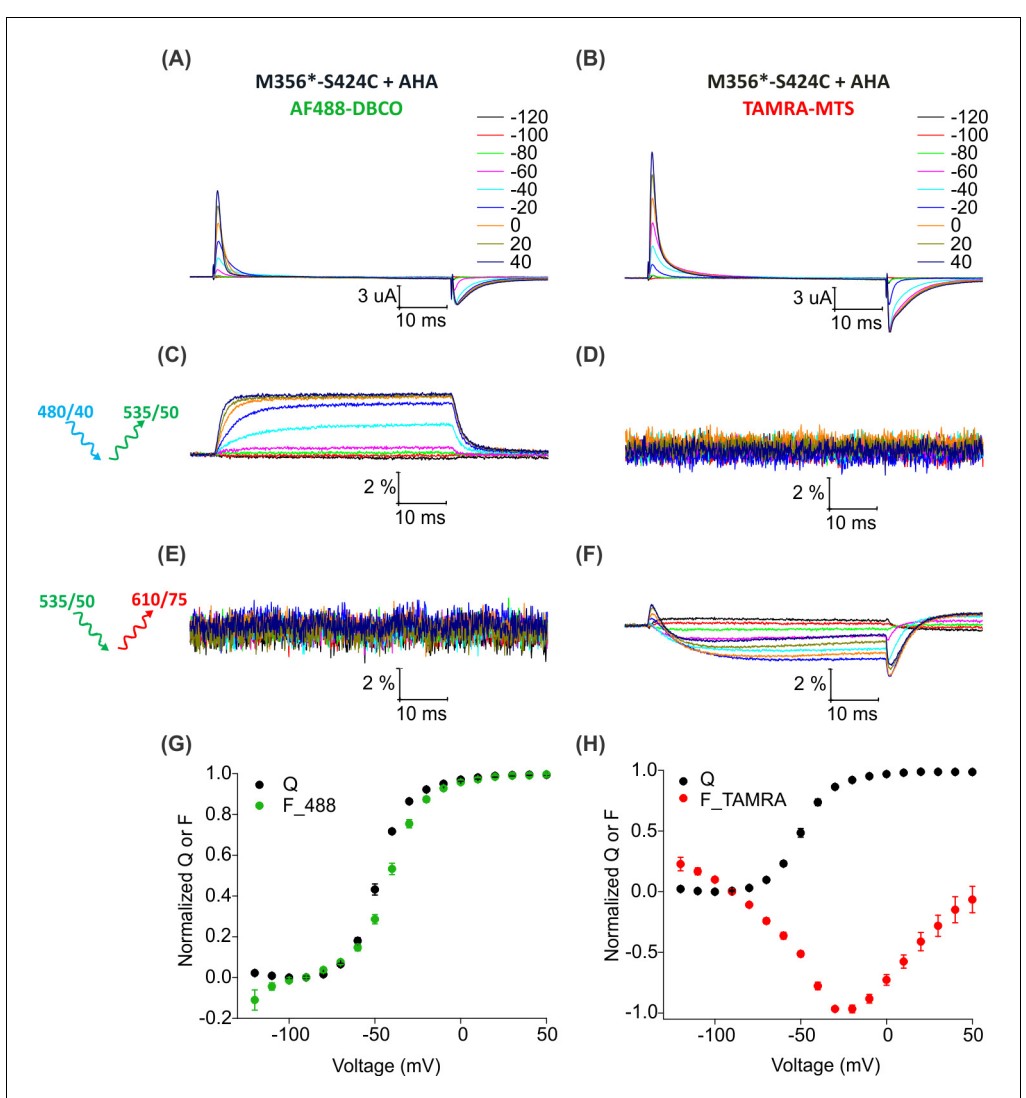

**Figure 6.** Fluorescence responses from the Shaker Kv channel labeled with azide or thiol-reactive fluorophores. (**A–B**) Gating currents from oocytes expressing Shaker-M356*-S424C (M356/M448L/C245V/C462A/S424C) in the presence of AHA and labeled with AF488-DBCO (**A**) or TAMRA-MTS (**B**). (**C–D**) Fluorescence responses from oocytes labeled with AF488-DBCO (**C**) or TAMRA-MTS (**D**) through 488 filter cube (ex. 480/40 nm; em. 535/50 nm). (**E–F**) Fluorescence response from oocytes labeled with AF488-DBCO (**E**) or TAMRA-MTS (**F**) through TAMRA filter cube (ex. 535/50 nm; em. 610/75 nm). (**G–H**) Q-V (Q, black) and steady state F-V relationships (F_488; green, F_TAMRA; red) obtained from oocytes labeled with AF488-DBCO (**G**) or TAMRA-MTS (**H**) fluorophores. All data points are the mean ± SEM, n = 5–7.

DOI: https://doi.org/10.7554/eLife.50776.012

For two-color labeling, oocytes expressing the Shaker Kv channel containing only the M356 or S424C sites were used to assess the degree of cross-reactivity between cyclooctyne and cysteine residues (*Beatty et al., 2010*; *Conte et al., 2011*; *van Geel et al., 2012*; *Zhang et al., 2018a*). To minimize the cross-reactivity, oocytes were labeled sequentially with TAMRA-MTS and then AF488-DBCO (See Materials and methods). As expected, two-color labeled Shaker-M356* produced voltage-dependent fluorescence changes through the AF488 filter cube (*Figure 7E,K*) which closely resembled Shaker-M356* labeled only with AF488-DBCO (*Figure 4A,C*), and there was no measurable change in fluorescence intensity through the TAMRA filter cube (*Figure 7H*). Similarly, two-color labeling of oocytes expressing Shaker-M356*-S424C in the absence of AHA showed no voltage-dependent fluorescence through the AF488 filter cube (*Figure 7F*), while the fluorescence response through the TAMRA filter cube was consistent with previous reports for S424C labeled with TAMRA-maleimide (*Figure 7I,L*) (*Claydon et al., 2007*). The lack of voltage-dependent fluorescence responses in the TAMRA channel (ex. 535/50 nm; em. 610/75 nm) after two-color labeling of M356* (*Figure 7H*) indicates that TAMRA-MTS does not cross-react with AHA at M356, since TAMRA produces a robust response when attached at this position using cysteine chemistry (*Cha and Bezanilla, 1997*; *Mannuzzu et al., 1996*). Similarly, the absence of voltage-dependent fluorescence responses in the AF488 channel for two-color labeling of M356*-S424C indicates that AF488-DBCO does not cross-react with S424C under these labeling conditions, since labeling S424C with AF488-C5-maleimide produces robust fluorescence responses (*Figure 7—figure supplement 1*). Finally, the two-color labeled Shaker-M356*-S424C showed distinct voltage-dependent fluorescence responses through both AF488 and TAMRA filter cubes (*Figure 7G,J,M*), similar to the single color labeling (*Figure 6*). These results demonstrate that AHA- and cysteine-mediated fluorescent labeling approaches can be combined for chemically selective and site-specific installation of different fluorophores into the Shaker Kv channel.

To determine whether individual Shaker Kv channels have been simultaneously labeled with both fluorophores, we looked for direct intramolecular energy transfer between AF488 and TAMRA. In the structure of the Kv1.2/2.1 paddle chimera (*Long et al., 2007*), the Cα distances between the residues corresponding to M356 in S4 and S424 in the four subunits forming the pore domain are 23.7 Å, 38.5 Å, 45.8 Å and 55 Å (*Figure 8—figure supplement 1*), near enough to allow FRET from AF488 to TAMRA (where $R_0$ ~55 Å). To distinguish such intra-molecular FRET in our system, we must account for background sources of fluorescence (e.g. fluorophores attached to other surface proteins, oocyte auto-fluorescence, etc.) as well as 'bleed through' due to direct emission of the donor and direct excitation of the acceptor through the FRET filter cube. Conveniently, the background fluorescence is independent of voltage, so its contribution can be excluded by considering only the voltage-dependent fluorescence changes (ΔF). *Figure 8* shows the voltage-dependent fluorescent changes for constructs containing only the fluorescent donor site (Shaker-M356*) or the fluorescent acceptor site, (Shaker M356A-S424C) or both donor and acceptor sites (Shaker M356*-S424C) in the presence of AHA. Because the signal through the FRET cube (Alexa 488 excitation: ex. 480/40 nm; TAMRA emission: em. 535/50 nm) includes contributions from AF488 fluorophores emitting directly into the TAMRA channel and TAMRA fluorophores directly excited by the AF488 excitation, we used oocytes expressing the Shaker-M356*-S424C labeled only with AF488-DBCO or TAMRA-MTS to estimate and correct for this spectral bleed-through (*Figure 8—figure supplement 2*). Subtraction of AF488-DBCO direct emission into the TAMRA channel should be quite reliable as the emission in the AF488 channel and TAMRA channel have identical voltage dependent behavior with a mean bleed through ratio of 0.142 ± 0.004 at 50 mV (*Figure 8—figure supplements 2G* and *3*). In contrast, the subtraction of the TAMRA direct excitation signal is more approximate because the voltage-dependence of TAMRA emission depends on the excitation wavelength and was not identical with AF488 and TAMRA excitation (*Figure 8—figure supplement 2H*). The mean bleed through ratio for TAMRA was estimated to be 0.076 ± 0.004 at 50 mV. (*Figure 8—figure supplement 3*). Nevertheless, both the raw and corrected FRET signals (Alexa 488 excitation; TAMRA emission) are larger when both donor and acceptor are present (*Figure 8L,O*) compared to when only the donor (*Figure 8J*) or acceptor (*Figure 8K*) is present. Furthermore, the increase in the FRET signal upon depolarization correlates with the upward movement of the S4 helix (and donor) towards the acceptor with the corrected FRET F-V relationship closely following the Q-V relationship for the oocytes expressing Shaker-M356*-S424C in the presence of AHA (*Figure 8R*). Collectively, these results

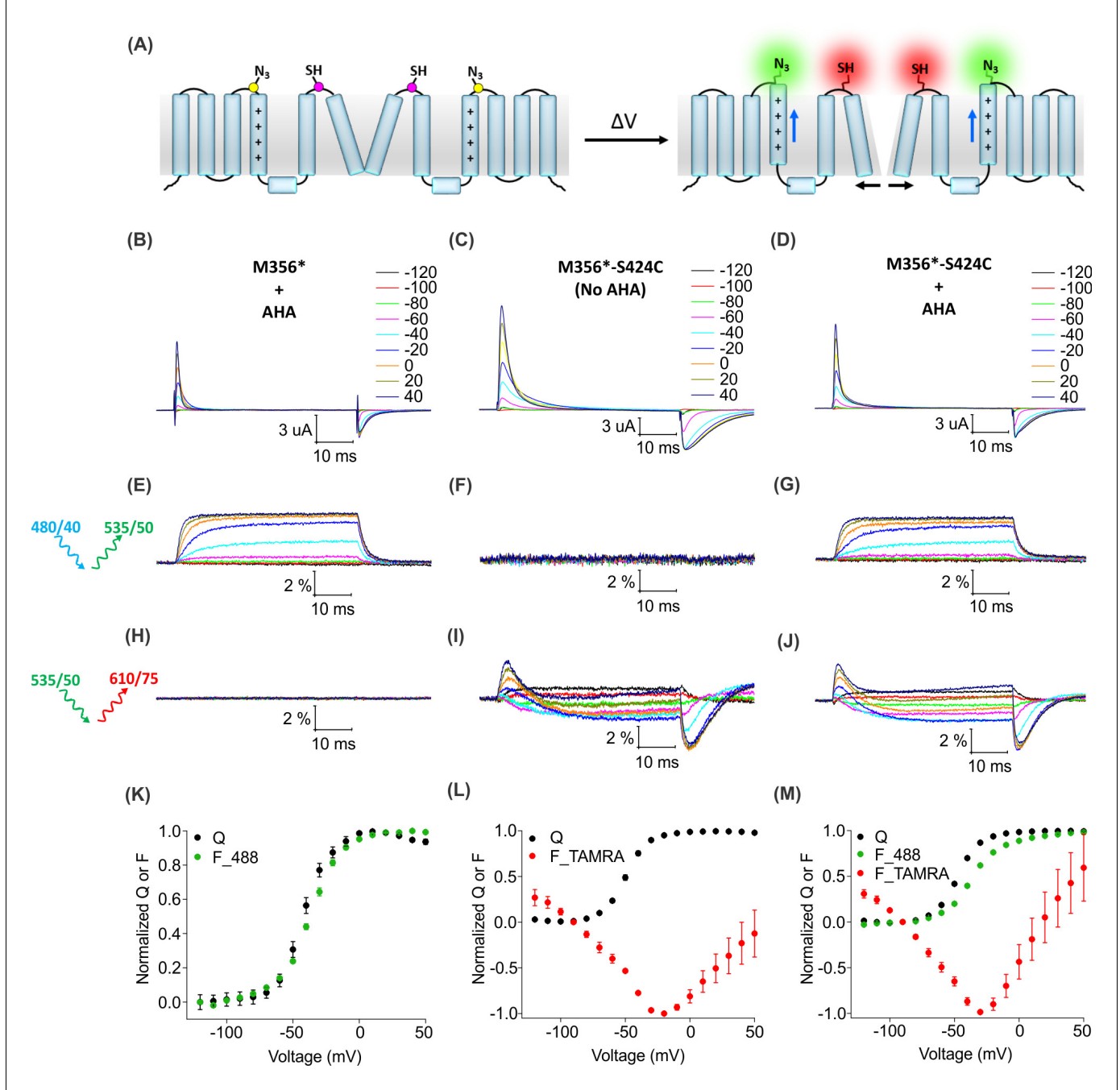

**Figure 7.** Two color labeling of the Shaker Kv channel through AHA and cysteine. (**A**) A schematic for two-color labeling of the Shaker Kv channel. Each subunit contains an azide group from AHA (yellow) on the top of S4 within the voltage-sensing domain and a thiol group from cysteine (magenta) in the pore domain. Voltage-dependent conformational changes in the channel (blue and black arrows) result into a change in the fluorescence intensity of AF488-DBCO (green) and TAMRA-MTS (red) fluorophores. (**B–D**) Gating currents obtained from two-color labeled oocytes expressing Shaker-M356* (M356/M448L/C245V/C462A) in the presence of AHA (**B**) or Shaker-M356*-S424C (M356/M448L/C245V/C462A/S424C) in the absence (**C**) or presence of AHA (**D**). (**E–J**) Fluorescence responses from the two-color labeled oocytes through 488 filter cube (ex. 480/40 nm; em. 535/50 nm) (**E–G**) and TAMRA filter cube (ex. 535/50 nm; em. 610/75 nm) (**H–J**). (**K–M**) Q-V (Q, black) and steady-state F-V relationships (F_488; green, F_TAMRA; red) obtained from oocytes labeled with both AF488-DBCO and TAMRA-MTS. In all cases, data points are the mean ± SEM (n = 4–6). For Shaker-M356*, maximal ΔF/F (%) for TAMRA filter cube was 0.013 ± 0.003 (**H**). For Shaker-M356*-S424C without AHA, maximal ΔF/F (%) for 488 filter cube was 0.046 ± 0.011 (**F**).

DOI: https://doi.org/10.7554/eLife.50776.013

The following figure supplement is available for figure 7:

*Figure 7 continued on next page*

*Figure 7 continued*

**Figure supplement 1.** Voltage-dependent fluorescence responses from Shaker-M356*-S424C after labeling with AF488-C5-Maleimide.
DOI: https://doi.org/10.7554/eLife.50776.014

establish that AHA incorporation and cysteine mutagenesis can be efficiently combined to carry out site-specific two-color labeling of individual Shaker Kv channels in live cells.

## Discussion

In the present study, we introduce a cysteine-independent method to engineer membrane proteins with crosslinkable chemical groups and subsequently modify them with spectroscopic probes using a bioorthogonal chemical reaction in live cells. We used the non-canonical amino acid, azidohomoalanine (AHA) (*Kiick et al., 2002*), to introduce azide groups in place of methionine residues in the Shaker Kv channel. Our results establish that AHA can be readily incorporated into the Shaker Kv channel in an efficient and residue-specific manner. Using SPAAC chemistry with azide-reactive cyclooctyne conjugated reagents (*Agard et al., 2004*), we demonstrate the utility of AHA incorporation for site-specific installation of fluorescent probes in the Shaker Kv channel to follow the conformational changes with voltage-clamp fluorometry in *Xenopus* oocytes (*Cha and Bezanilla, 1997*; *Mannuzzu et al., 1996*). We were able to combine AHA and cysteine-mediated fluorescent labeling for simultaneous labeling with two different fluorophores at specific sites in the voltage-sensing and pore domains of the Shaker Kv channel. We also demonstrate that a voltage-dependent FRET response can be detected with the two-color labeled Shaker Kv channel exclusively when unique cysteine and AHA-substituted methionine residues are both present. Taken together, our results suggest that AHA incorporation and cysteine mutagenesis provide a straightforward and robust way of incorporating two distinct reactive groups into the Shaker Kv channel expressed in live cells.

We believe this approach will work for other membrane proteins, but several important issues should be considered for each potential application. One of the most important considerations is the number and location of methionine residues in the protein of interest. Methionine residues have similar abundance as cysteine residues in membrane proteins and they tend to be located towards the center of the lipid bilayer (*Koehler Leman et al., 2018*), suggesting that our AHA-based approach will have similar applications and limitations when compared to cysteine-based approaches. Although methionine residues are somewhat more abundant in other membrane proteins compared to the Shaker Kv channel, their prevalence is comparable to cysteine residues within regions potentially accessible to the extracellular solution (*Supplementary file 3*). Importantly, not all methionine residues in the extracellular half of the protein will be accessible to azide-reactive alkyne probes and would need to be removed. In the Shaker Kv channel, for example, out of the three methionine residues in the extracellular half of the protein, while M356 can be robustly labeled with DBCO-biotin in the presence of AHA, M448 exhibits barely detectable labeling (data not shown) and M440 is inaccessible, as seen by the absence of streptavidin pulldown for the M356A/M448L double mutant (*Figure 1F*). Moreover, it has been previously demonstrated that site-specific fluorescence responses can be measured using cysteine-based approaches without removing the native cysteine residues (*Savalli et al., 2006*). Thus, using AHA for fluorescent labeling on the extracellular side of membrane proteins should be generally applicable, even for some of the larger membrane proteins we analyzed (*Supplementary file 3*).

The degree to which replacement of methionine with AHA perturbs the functional properties of a membrane protein is another important consideration in applying the approaches described here. In our experience with the Shaker Kv channel, AHA incorporation is well-tolerated, showing minimal perturbations in cellular expression level (*Figure 1*), voltage-dependent gating properties (*Figure 2*) and response of the channel to a gating modifier toxin, GxTx1E (*Figure 2—figure supplement 1*). This can be attributed to the highly isosteric nature of methionine and AHA residues (*Kiick et al., 2002*), making it a suitable substrate for endogenous protein translation machinery of *Xenopus* oocytes and precluding substantial changes to the allosteric transitions required for voltage-dependent activation and deactivation of the channel. In most circumstances, it is likely that replacement of methionine with AHA will be well tolerated, although the effect of substituting a new methionine residue will depend on the identity of that specific site. It is also important to consider that the

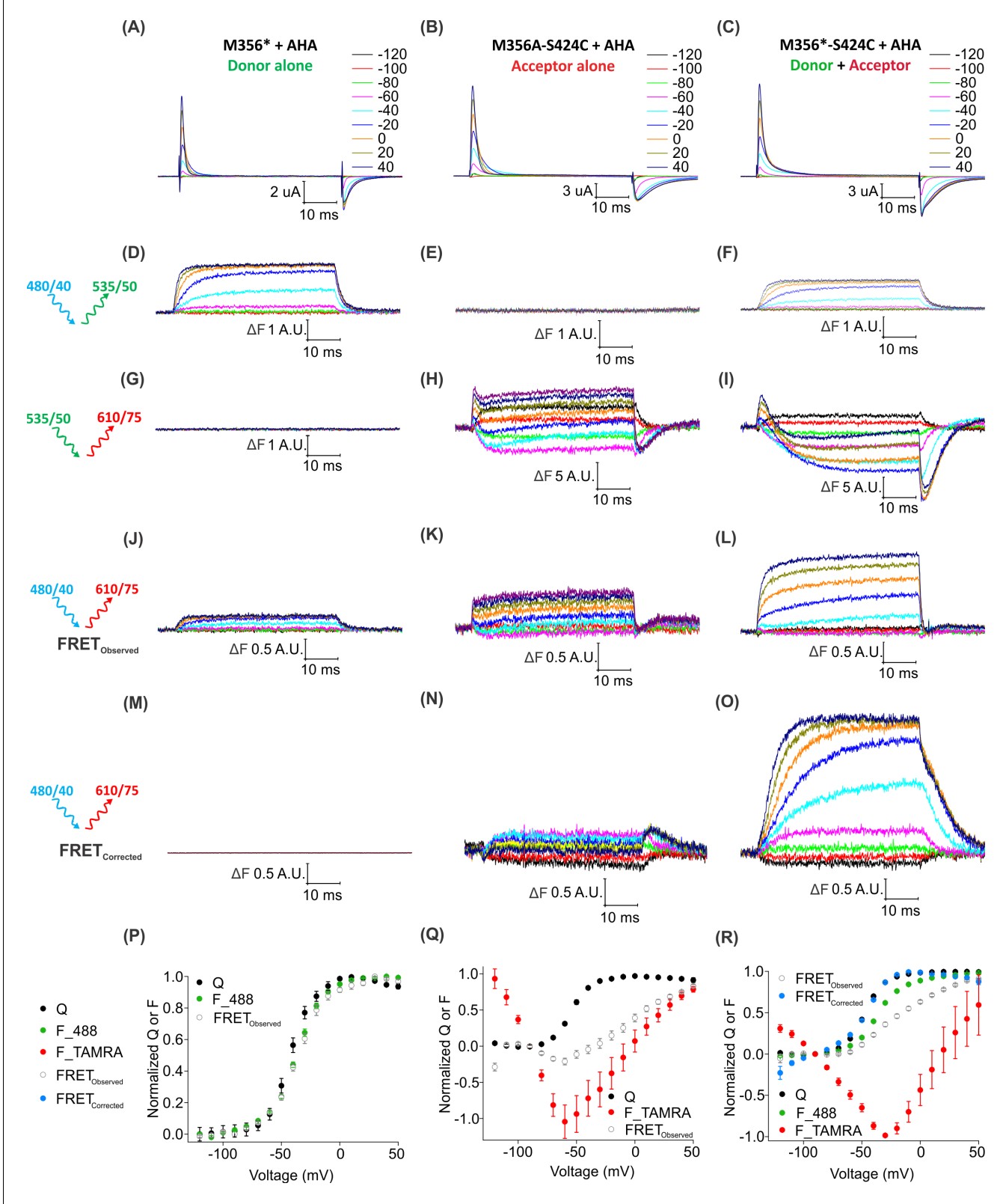

**Figure 8.** Two-color labeling through AHA and cysteine installs different fluorophores within the Shaker Kv channel. (A–C) Gating currents obtained from two-color labeled oocytes expressing Shaker-M356* (M356/M448L/C245V/C462A) (A), Shaker-M356A-S424C (M356A/M448L/C245V/C462A/S424C) (B) or Shaker-M356*-S424C (M356/M448L/C245V/C462A/S424C) (C) in the presence of AHA. (D–O) Fluorescence responses from two-color labeled oocytes through 488 filter cube (ex. 480/40 nm; em. 535/50 nm) (D–F), TAMRA filter cube (ex. 535/50 nm; em. 610/75 nm) (G–I) or FRET filter cube (ex.

*Figure 8 continued on next page*

*Figure 8 continued*

480/40 nm; em. 610/75 nm) before (**J–L**) and after (**M–O**) bleed through correction. (**P–R**) Q-V (Q, black) and steady-state F-V relationships (F_488; green, F_TAMRA; red, FRET$_{Observed}$; gray and FRET$_{Corrected}$; cyan) obtained from oocytes labeled with both AF488-DBCO and TAMRA-MTS. In all cases, data points are the mean ± SEM (n = 5–7). For Shaker-M356*, maximal ΔF for TAMRA filter cube was 0.131 ± 0.003 (**G**) and FRET$_{Corrected}$ was −0.00585 ± 0.004 (**M**). For Shaker-M356A-S424C, maximal ΔF for 488 filter cube was 0.010 ± 0.008 (**E**) and FRET$_{Corrected}$ cube was 0.032 ± 0.018 (**N**).

DOI: https://doi.org/10.7554/eLife.50776.015

The following figure supplements are available for figure 8:

**Figure supplement 1.** Structure of Kv1.2–2.1 paddle chimera (2R9R) indicating the Cα distances between residues corresponding to M356 and S424 in the Shaker Kv channel (*Long et al., 2007*).

DOI: https://doi.org/10.7554/eLife.50776.016

**Figure supplement 2.** Bleed through of fluorescence from AF488 and TAMRA through FRET cube.

DOI: https://doi.org/10.7554/eLife.50776.017

**Figure supplement 3.** Bleed through ratio of fluorescence from AF488 and TAMRA through FRET cube.

DOI: https://doi.org/10.7554/eLife.50776.018

relatively bulky DBCO group we used in this study and the product of SPAAC reaction with AHA may not be well-accommodated at all positions in a protein of interest. However, in our experience thus far, labeling the voltage-sensing domain of the Shaker Kv channel with DBCO-conjugated fluorophores does not dramatically alter voltage sensor activation (*Figures 3C*, *4E and G*). We also note that smaller azide-reactive cyclooctyne reagents like the bicyclononynes (BCN) group have also been conjugated to fluorophores and are commercially available for fluorescent labeling of AHA-modified membrane proteins (*Supplementary file 2*) (*Dommerholt et al., 2010*; *Dommerholt et al., 2014*; *Leunissen et al., 2014*).

The extent of AHA incorporation will also be an important variable to consider in planning future applications of our methods and is clearly a limitation in comparison to cysteine mutagenesis. Although the pull-down observed with the azide-reactive DBCO-sulfo-biotin is comparable to that with the amine-reactive NHS-sulfo-biotin (*Figure 1E*, lane 2 and 4 and *Figure 1G*), suggesting that the extent of AHA incorporation is robust, this assay is relatively qualitative. We are currently developing quantitative methods to measure the extent of AHA incorporation using mass spectrometry and/or azide-reactive cyclooctyne polyethylene glycol (PEG) polymers (*Darabedian et al., 2018*). Background labeling of cysteine and methionine residues present within endogenously expressed proteins might also prevent detection of distinct fluorescent signals from proteins that do not express to high enough levels. However, it is important to appreciate that specifically detecting the stimulus-dependent changes in fluorescence may be sufficient to distinguish responses originating from the protein of interest. For instance, we did not observe any voltage-dependent changes in fluorescence with the Shaker-M356A construct in the presence of AHA (*Figure 3E*), even though background labeling of other proteins was clearly detectable in the presence of AHA (*Figure 3B*). Nevertheless, limitations imposed by partial AHA incorporation and incomplete SPAAC reactions combine to produce only 2-fold smaller fluorescence responses for our AHA-based labeling approach when compared to cysteine-based methods (*Figure 5*), providing a convenient benchmark to suggest that AHA-based methods should be viable for most proteins that have already been successfully studied using cysteine-based approaches (*Supplementary file 3*) (*Cowgill and Chanda, 2019*; *Horne and Fedida, 2009*; *Priest and Bezanilla, 2015*).

Finally, it is important to keep in mind that the array of DBCO- or BCN-conjugated fluorophores which are commercially available is currently more limited (*Supplementary files 1* and *2*) compared to the thiol-reactive probes available for cysteine-based approaches. Nevertheless, there are certain DBCO-conjugated fluorophores (e.g. AF555-DBCO and AF594-DBCO) that may have better properties than AF488-DBCO because they absorb far from the UV region, and thus the fluorescence responses should have less contamination from endogenous oocyte fluorescence and improved signal/noise ratios. Our preliminary results with AF555-DBCO and AF594-DBCO show that these spectrally distinct fluorophores also exhibit voltage-dependent changes in fluorescence when installed at M356 using AHA (data not shown). The ability to install different azide-reactive fluorescent reporters through AHA incorporation into a membrane protein will hopefully provide motivation to generate better reagents with optimal spectral properties and linkers between the reactive group and fluorophore.

An important aspect of combining AHA incorporation and cysteine mutagenesis is the compatibility between azide and thiol-mediated chemical reactions for installing two different biophysical probes into a single protein. It is important to mention that some cyclooctyne groups can react with cysteine residues (*Beatty et al., 2010*; *Conte et al., 2011*; *van Geel et al., 2012*; *Zhang et al., 2018a*), however, our approach of first labeling cysteine residues with thiol-reactive probes works efficiently for achieving site-specific fluorescence responses from azide and thiol-reactive fluorophores with negligible cross-reactivity between cyclooctyne and cysteine residues (*Figures 7* and *8*). Given that the reaction rates between cysteine and MTS/maleimide reagents are considerably faster ($\sim 10^3$–$10^4$ M$^{-1}$s$^{-1}$) (*Saito et al., 2015*) than the reaction between AHA and cyclooctynes (0.1–1 M$^{-1}$s$^{-1}$) (*Dommerholt et al., 2016*; *Lang and Chin, 2014*), it should be possible to achieve specific labeling when the two reactions are carried out at the same time. In addition, other azide-containing amino acids such as p-azidophenylalanine have been widely incorporated into membrane proteins through nonsense suppression methods (*Daggett and Sakmar, 2011*; *Rannversson et al., 2016*; *Zhu et al., 2014*), and thus can also be combined with cysteine mutagenesis similar to what we have shown here with AHA incorporation and cysteine mutagenesis (*Figure 7*).

Our FRET measurements demonstrate that we can simultaneously install two different fluorophores site-specifically into the Shaker Kv channel using a combination of AHA incorporation and cysteine mutagenesis (*Figure 8*). Recently, the fluorescent non-canonical amino acid, Anap, has been incorporated and combined with cysteine mutagenesis to install two fluorescent reporters into the Shaker Kv channel and simultaneously monitor conformational changes on the intracellular and extracellular side of the protein (*Kalstrup and Blunck, 2013*). Anap incorporation has also been combined with TETAC, a cysteine-reactive transition metal binding cyclen, for measuring intramolecular distances using tmFRET (*Dai et al., 2019*). Although Anap incorporation is highly site-specific and allows quantitative estimation of short distances (10–20 Å), its incorporation diminishes protein expression considerably (*Aman et al., 2016*; *Shandell et al., 2019*; *Zagotta et al., 2016*). Moreover, Anap requires UV excitation, and thus suffers from contamination with cellular autofluorescence in some cell types (*Chatterjee et al., 2013*). The combination of AHA incorporation and cysteine mutagenesis would provide flexibility to choose the donor and acceptor pairs for measuring a wide range of distances and it would be particularly exciting to carry out FRET measurements between fluorophores installed through AHA and TETAC installed with cysteine. In addition to two-color labeling, combining AHA incorporation and cysteine mutagenesis would be valuable for an array of other biophysical applications. Installation of fluorophores with AHA effectively frees up cysteine mutagenesis, providing an opportunity to spectroscopically monitor the impact of cysteine modification, or disulfide/metal bridge formation on the structural rearrangements of membrane proteins. It would also be exciting to combine AHA incorporation and cysteine mutagenesis for bioorthogonal installation of electron paramagnetic probes at two independent sites, greatly expanding the types of distance measurements that could be achieved. Finally, due to inherent orthogonality between AHA incorporation, cysteine mutagenesis and nonsense suppression methods, it is conceivable that three independent biophysical reporters can be installed within a protein to further constrain FRET based measurements and/or study multiple conformational changes simultaneously. Given that AHA incorporation has been widely used in a variety of different cell types (*Dieterich et al., 2010*; *Dieterich et al., 2006*; *Erdmann et al., 2015*; *Glenn et al., 2017*; *Hinz et al., 2012*; *Link et al., 2004*; *Ma and Yates, 2018*), the approach described here with *Xenopus laevis* oocytes should be readily applicable to studies in other cellular expression systems.

## Materials and methods

**Key resources table**

| Reagent type (species) or resource | Designation | Source or reference | Identifiers | Additional information |
|---|---|---|---|---|
| Gene (*Drosophila melanogaster*) | Shaker | Gene ID: 32780 | | |

*Continued on next page*

*Continued*

| Reagent type (species) or resource | Designation | Source or reference | Identifiers | Additional information |
|---|---|---|---|---|
| Oocytes (*Xenopus laevis*, female) | *Xenopus* Oocytes or oocytes | Xenopus Express | IMP XL FM | |
| Antibody | Anti-myc antibody | ThermoFisher | Cat. No. 46–0603 | 1:1000 (2 µl in 2 ml) |
| Antibody | HRP-conjugated anti-mouse secondary antibody | Amersham ECL | Cat. No. NA931VS | 1:3750 (4 µl in 15 ml) |
| Recombinant DNA reagent | pGEMHE vector | | | *Liman et al., 1992* |
| Peptide, recombinant protein | GxTx1E | This paper | Toxin | |
| Chemical compound, drug | NHS-sulfo-biotin | ThermoFisher | Cat. No. 21335 | 1 mM from 10X stock in $ddH_2O$ |
| Chemical compound, drug | DBCO-sulfo-biotin | Sigma | Cat. No. 760706 | 1 mM from 10X stock in $ddH_2O$ |
| Chemical compound, drug | AF488-DBCO | Click Chemistry Tools | Cat. No. 1278 | 100 µM from 100X stock in anhydrous DMSO |
| Chemical compound, drug | AF488-C5-Maleimide | Click Chemistry Tools | Cat. No. 1014 | 100 µM from 100X stock in anhydrous DMSO |
| Chemical compound, drug | TAMRA-MTS | Toronto Research Chemicals | Cat. No. T305175 | 10 µM from 1000X stock in $ddH_2O$ |
| Chemical compound, drug | Azidohomoalanine (AHA) | Bachem | Cat. No. F-4265 | 4 mM from 100 mM stock in $ddH_2O$ |

## Molecular biology

All the constructs were generated in the pGEMHE vector (*Liman et al., 1992*) with Shaker-IR (ΔN, 6–46) (*Hoshi et al., 1990*) as the common background. The mutations for methionine and cysteine residues were carried out using the QuickChange Lightning site-directed mutagenesis kit as per manufacturer's protocol (Agilent Technologies). The DNA sequence of all constructs and mutants was confirmed by automated DNA sequencing and complementary RNA (cRNA) was synthesized using T7 polymerase after linearizing the DNA with *NheI* restriction enzyme. The RNA was purified using the RNAse easy kit (Qiagen), eluted in RNAse-free water and stored at −80°C until use. All the chemicals were purchased from Sigma-Aldrich unless specified.

## Detection of AHA incorporation through surface biotinylation of *Xenopus laevis* oocytes

Female *Xenopus laevis* animals were housed and surgery was performed according to the guidelines of the National Institutes of Health, Office of Animal Care and Use (OACU) (Protocol Number 1253–18). Oocytes were removed surgically and incubated with agitation for 1 hr in a solution containing (in mM) 82.5 NaCl, 2.5 KCl, 1 $MgCl_2$, 5 HEPES, pH 7.6 (with NaOH), and collagenase (2 mg/ml; Worthington Biochemical, Lakewood, NJ). All surface biotinylation experiments were carried out with ShakerΔ5-V478W-myc (*Hackos et al., 2002*; *Milescu et al., 2013*) containing a myc tag at the C-terminal. Defolliculated oocytes were injected with 50 nl of channel RNA (~500 ng/µl) after preincubating them in the absence or presence of 4 mM AHA (Bachem, 100 mM stock in $ddH_2O$) at 17°C overnight in an ND96 oocyte maintenance buffer, containing (in mM): 96 NaCl, 2 KCl, 5 HEPES, 1 $MgCl_2$ and 1.8 $CaCl_2$ plus 50 mg/ml gentamycin, pH 7.6 with NaOH. After four days of maintaining the oocytes at 17°C, excess AHA was removed with 5–6 washes of ND96 and oocytes were labeled with amine or azide reactive biotin reagents, EZ-Link-sulfo-NHS-LC-biotin (ThermoFisher) or DBCO-sulfo-biotin (Sigma), according to the previously published protocol with minor modifications

(*Silberberg et al., 2005*; *Zhang et al., 2018b*). Twenty healthy oocytes were incubated with 1 mM of each biotin probe (10X stock in ddH$_2$O) in separate wells of a 24-well plate in a final volume of 0.5 ml at room temperature. The reaction was terminated after 20 min for NHS-sulfo-LC-biotin and 60 min for DBCO-sulfo-biotin by transferring the oocytes to ND96, followed by 6–8 washes to remove excess biotinylation reagent. Subsequently, oocytes were homogenized in 400 µl of lysis buffer containing (in mM): 100 NaCl, 20 Tris·Cl, pH 7.4, 1% Triton X-100, 5 µl/ml protease inhibitor mixture (Sigma). Homogenization and all subsequent steps were performed at 4°C. After centrifugation at 16,000 × $g$ for 3 min, a 20 µl aliquot of the supernatant (total cell protein) was mixed with equal volume of 2 × NuPAGE LDS sample buffer (ThermoFisher) plus reducing agent: 50% 4 × LDS sample buffer (Bio-Rad), 10% 2-mercaptoethanol and 40% 100 mM DTT. The remaining supernatant was diluted 1:1 with the lysis buffer and 60 µl of High Capacity NeutrAvidin agarose beads (Thermo-Fisher) were added followed by gentle tumbling overnight at 4°C. The NeutrAvidin agarose beads were washed six times with the lysis buffer with a 2 min centrifugation (16,000 × $g$) step between each wash. At the end of the final wash, 40 µl of 2 × LDS sample buffer plus reducing agent was added to the beads and samples were heated at 70°C for 10 min. Following a 2 min centrifugation (16,000 × $g$), the supernatant (surface protein) and total cell protein (collected earlier) were separated in 10% Bis-Tris acrylamide gel (ThermoFisher) using a MOPS running buffer containing (in mM): 20 Tris base, 20 MOPS, 1.25 EDTA, 0.1% SDS, pH 7.6. Seeblue Plus2 prestained ladder (ThermoFisher) was used as the protein molecular weight marker. After SDS-PAGE, proteins in the gel were transferred to nitrocellulose membrane using the iBLOT semi-dry transfer apparatus (Thermo-Fisher). The nitrocellulose membrane was probed with mouse anti-myc antibody (ThermoFisher, Cat. No. 46–0603) diluted 1:1000 in TBS-T containing (in mM): 25 Tris, 137 NaCl, 3 KCl, 0.05% Tween20 followed by HRP-conjugated anti-mouse secondary antibody (4 µl in 15 ml TBS-T). The blot was developed using Immobilon ECL Western detection reagents (Millipore). Densitometry was performed with the Image lab software (Bio Rad).

## Two-electrode voltage clamp recordings of macroscopic ionic currents

All ionic currents were recorded using the Shaker-IR construct where residues 6–46 were deleted to remove N-type inactivation (*Hoshi et al., 1990*). For experiments with the tarantula toxin GxTx1E, the toxin-sensitive ShakerΔ5 construct (L327I, A328F, V330T, V331E and A332S) was used (*Milescu et al., 2013*). GxTx1E toxin was synthesized on an ABI peptide synthesizer using Fmoc chemistry, refolded in vitro and purified as previously described (*Gupta et al., 2015*). Experiments with AHA-modified channel were performed after preincubating the oocytes in 4 mM AHA (prepared in ND96 from a 100 mM stock in ddH$_2$O) overnight, followed by cRNA injection. Oocytes were injected with 50 nl of channel RNA (5–10 ng/µl) in the absence or presence of AHA and studied after 1–4 days to allow for expression at 17°C in ND96 solution. All the recordings were performed using the two-electrode voltage-clamp recording technique (OC-725C amplifier; Warner Instruments, Hamden, CT) using a 150 µl recording chamber. Data were filtered at 1 kHz and digitized at 5–10 kHz using Digidata 1321A interface board and pCLAMP 10 software (Molecular Devices, Sunnyvale, CA). Microelectrode resistances were 0.2–0.8 MΩ when filled with 3 M KCl. The external recording solution contained (in mM): 50 KCl, 50 NaCl, 10 HEPES, pH 7.6 with NaOH at room temperature (~22°C).

## Voltage-clamp fluorometry

All voltage-clamp fluorometry experiments were performed using the non-conducting V478W mutant of the Shaker Kv channel (*Hackos et al., 2002*; *Kitaguchi et al., 2004*), with additional methionine or cysteine mutations as indicated in the text and figure legends. Oocytes were injected with 50 nl of channel RNA (100–500 ng/µl) in the absence or presence of 4 mM AHA and maintained at 17°C for 1–5 days. Fluorescent labeling of oocytes was carried out by first removing excess AHA with 5–6 washes with ND96, followed by incubation with 100 µM AF488-DBCO (Alexa fluorophore 488-dibenzocyclooctyne, Click Chemistry Tools, 100X stock in DMSO) or AF488-C5-maleimide (Alexa fluorophore 488-C5-maleimide, Click Chemistry Tools, 100X stock in DMSO) for 60 min at room temperature in 0.5 ml ND96. Oocytes were transferred to fresh ND96 and washed five times (5 min each) to remove the excess fluorophore and stored in the dark at 10–13°C prior to performing experiments. For TAMRA-MTS (2-((5 (6)-tetramethylrhodamine)carboxylamino)ethyl

methanethiosulfonate; Toronto Research Chemicals), oocytes were incubated with 10 μM of the fluorophore (1000X stock in ddH2O) in ND96 at 4°C for 60 min, followed by five washes. For the two-color labeling, oocytes were first labeled with TAMRA-MTS, followed by AF488-DBCO labeling as documented above. Two-electrode voltage clamp recordings were obtained using a Dagan CA-1B amplifier. Electrodes were filled with 3M KCl and had resistances between 0.2–0.8 MΩ. The external recording solution was ND96 without gentamycin, containing (in mM): 96 NaCl, 2 KCl, 5 HEPES, 1 MgCl$_2$ and 1.8 CaCl$_2$, pH 7.6 with NaOH. For all gating current measurements, Q was obtained by integrating the OFF gating current elicited by repolarization to the holding voltage. Fluorescence signals were acquired through a 40X, 0.8-NA objective (Olympus LUMplanFLN) on an Olympus BX51WI microscope and a photodiode. For Alexa 488 signals, excitation filter, emission filter and dichroic were ET480/40, ET535/50 and T510nm, respectively (Chroma Tech.). For TAMRA-MTS, excitation filter, emission filter and dichroic were HQ535/50, HQ610/75 and T570pxrxt, respectively (Chroma Tech.). For the FRET cube, the excitation filter, emission filter and dichroic were ET480/40, HQ610/75 and T570pxrxt, respectively (Chroma Tech.). The signal from the photodiode was low-pass filtered at 3 kHz and sampled at 20 kHz through a Digidata-1440A controlled by pClamp10 (Molecular Devices). The light source used for the illumination was either a blue (470/24 nm) or a green (550/15 nm) LED (Lumencor, Spectra X). The bleed through fluorescence ratio for AF488 and TAMRA were calculated by using single-color labeled oocytes expressing Shaker-M356*-S424C in the presence of AHA and normalizing the magnitude of the steady-state voltage-dependent change in fluorescence intensity (ΔF, A.U.) obtained through the FRET cube by the one obtained from 488 and TAMRA filter cubes, respectively. All the fluorescence traces represent single recordings without averaging.

## Data and materials availability

All data needed to evaluate the conclusions in this paper are available in the main text and supplementary materials.

## Acknowledgements

We thank Anirban Banerjee, Joe Mindell, Miguel Holmgren, Mark Mayer, Yan Li and members of the Swartz laboratory for helpful discussions. We also thank Helena Chang for help with the surface biotinylation experiments and Tamas Lajtos for assistance with molecular biology.

## Additional information

### Competing interests

Kenton J Swartz: Reviewing editor, *eLife*. The other authors declare that no competing interests exist.

### Funding

| Funder | Author |
| --- | --- |
| National Institute of Neurological Disorders and Stroke | Kenton J Swartz |

The funders had no role in study design, data collection and interpretation, or the decision to submit the work for publication.

### Author contributions

Kanchan Gupta, Conceptualization, Data curation, Formal analysis, Investigation, Methodology, Resources, Validation, Writing—original draft, Writing—review and editing; Gilman ES Toombes, Conceptualization, Data curation, Formal analysis, Investigation, Methodology, Validation, Writing—original draft, Writing—review and editing; Kenton J Swartz, Conceptualization, Resources, Supervision, Funding acquisition, Validation, Investigation, Methodology, Project administration, Writing—review and editing

## Author ORCIDs

Kanchan Gupta https://orcid.org/0000-0002-8411-881X
Gilman ES Toombes https://orcid.org/0000-0001-8346-1790
Kenton J Swartz https://orcid.org/0000-0003-3419-0765

## Ethics

Animal experimentation: Female *Xenopus laevis* animals were housed and surgery was performed according to the guidelines of the National Institutes of Health, Office of Animal Care and Use (OACU) (Protocol Number 1253-18).

## Decision letter and Author response

Decision letter https://doi.org/10.7554/eLife.50776.024
Author response https://doi.org/10.7554/eLife.50776.025

# Additional files

## Supplementary files

• Supplementary file 1. Commercially available dibenzocyclooctyne (DBCO)-conjugated fluorophores https://clickchemistrytools.com/product-category/fluorescent-dyes/cu-free-click-chemistry/ - All these fluorophores are charged and membrane impermeable.
DOI: https://doi.org/10.7554/eLife.50776.019

• Supplementary file 2. Commercially available bicyclononyne (BCN)–conjugated fluorphores https://biotium.com/product/cf-dye-bcn/ - All these fluorophores are membrane impermeable.
DOI: https://doi.org/10.7554/eLife.50776.020

• Supplementary file 3. Prevalence and location of methionine and cysteine residues in a selection of membrane proteins previously studied with voltage-clamp fluorometry.
DOI: https://doi.org/10.7554/eLife.50776.021

• Transparent reporting form DOI: https://doi.org/10.7554/eLife.50776.022

## Data availability

All data generated or analysed during this study are included in the manuscript and supporting files.

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
