## [Decision Letter]

Thank you for submitting your article "Exploring structural dynamics of membrane proteins by combining bioorthogonal chemistry and cysteine mutagenesis" for consideration by *eLife*. Your article has been reviewed by three peer reviewers, and the evaluation has been overseen by a Reviewing Editor and Richard Aldrich as the Senior Editor. The following individual involved in review of your submission has agreed to reveal their identity: Stephan A Pless (Reviewer #1).

The reviewers have discussed the reviews with one another and the Reviewing Editor has drafted this decision to help you prepare a revised submission.

Summary:

In this manuscript, Gupta and colleagues present a new approach to study conformational changes of a voltage-gated ion channel by inserting two distinct fluorophores, one via a SPAAC-based method and one via a conventional, cysteine-based approach. This allows the simultaneous incorporation of two different fluorophores into the protein and the tracking of fluorescence changes originating from two distinct regions. The manuscript is very well written and the work is technically meticulously executed. The method chosen to incorporate AHA is straightforward and has potential to be applied to other proteins. Furthermore, the control experiments provide convincing evidence in support of the tolerability and feasibility of AHA incorporation into membrane proteins. Overall, the work has the potential to form the foundation for a valuable new experimental angle for ion channels. All the reviewers also agree that the manuscript will benefit from a more balanced presentation highlighting the possible limitations of the approach and by providing a thorough discussion of the FRET data.

Essential revisions:

1) The applicability of AHA in the study of structural dynamics should be discussed more extensively. While the incorporation of AHA is technically relatively simple and is a valuable addition to the means of embedding fluorescent dyes in membrane proteins, its general applicability will likely be somewhat compromised by the lack of site specificity, since targeting a methionine residue of interest would require other potentially modifiable methionine residues to be removed first. Despite the low abundance of methionine in membrane proteins, this number will still be notable in larger proteins (e.g., BK channels, NaVs and CaVs), which inevitably adds another layer of complexity to experimental execution and/or data interpretation. Moreover, it is reasonable to anticipate that the bulkiness of the product of the SPAAC-based fluorophore conjugation will not be as well tolerated as more compact non-canonical amino acids or fluorescent dyes incorporated using conventional methods due to the greater steric hindrance. This, together with its limitation to solvent-accessible sites, will likely put constraints on the choice of incorporation site and should be mentioned. Recognizing and discussing the potential issues associated with the broader application of this approach will strengthen the manuscript.

2) Related to the above point, I would ask for a discussion of the drawbacks to the method, such as the fact that a protein of interest may contain multiple accessible cysteine and methionine residues, or that background labeling of cysteines and methionines in other proteins may make the approach intractable in cases where the protein of interest does not express in sufficient quantities similar to Kv expression in oocytes.

3) The authors extensively describe the measurement and correction of the observed FRET signals, but the purpose of this experiment was not evident. If the goal was to measure distances of donor and acceptor, it is necessary to calculate distances/distance changes, and to compare the accuracy of the chosen method as a molecular ruler to that of other approaches. As it currently stands, the significance of the FRET experiments in this study is unclear. In fact, it is not obvious if the pair of fluorophores would even be suitable for measuring distances, given the direct excitation of the acceptor. Furthermore, the authors should discuss the implications of the slow and thus likely incomplete/asymmetric labelling via the SPAAC-based method in more detail, especially because their measurements can report on more than one fluorophore pair (as shown in Figure 8—figure supplement 1).

4) Comparing to the cysteine based method the AHA method shows lower efficiency of introducing fluorescent probes (Figure 5C, D). The authors pointed out this result in the text. However, it may be more helpful if more details of how this relatively low efficiency affected experimental processes can be described and troubleshooting procedures are discussed. A discussion of other precautions and possible pitfalls in using this method may be also helpful.

5) As pointed out by the authors, unnatural amino acid Anap has been used in studying ion channels, but it is known in the field that not all ion channels are suitable for the Anap approach. It will be nice to show that the AHA method can be used in some channels other than Shaker. Nevertheless, it may be out of the scope of this manuscript, and the ion channel field will find out the answer after the publication of this manuscript. We suggest that the title is modified to "Exploring structural dynamics of a membrane protein by…".

6) Combining click chemistry and cysteine labeling approaches for dual color labeling as needed for FRET has been described before (e.g. https://www.ncbi.nlm.nih.gov/pubmed/23317903). That said, I do think the paper would be of interest to the ion channel community in general, many of whom may not be familiar with these approaches. Especially the use of AHA to substitute methionine residues for later click labeling, which I suspect is less appreciated in the ion channel field. However, I would ask the authors to generally tone down their statements regarding the novelty of the approach.

---

## [Author Response]

Essential revisions:1) The applicability of AHA in the study of structural dynamics should be discussed more extensively. While the incorporation of AHA is technically relatively simple and is a valuable addition to the means of embedding fluorescent dyes in membrane proteins, its general applicability will likely be somewhat compromised by the lack of site specificity, since targeting a methionine residue of interest would require other potentially modifiable methionine residues to be removed first. Despite the low abundance of methionine in membrane proteins, this number will still be notable in larger proteins (e.g., BK channels, NaVs and CaVs), which inevitably adds another layer of complexity to experimental execution and/or data interpretation. Moreover, it is reasonable to anticipate that the bulkiness of the product of the SPAAC-based fluorophore conjugation will not be as well tolerated as more compact non-canonical amino acids or fluorescent dyes incorporated using conventional methods due to the greater steric hindrance. This, together with its limitation to solvent-accessible sites, will likely put constraints on the choice of incorporation site and should be mentioned. Recognizing and discussing the potential issues associated with the broader application of this approach will strengthen the manuscript.

We are quite optimistic about the application of our approach to other proteins, but we completely agree that an expanded discussion of potential limitations could be valuable to readers. The following text can be found in the Discussion:

“We believe this approach will work for other membrane proteins, but several important issues should be considered for each potential application. […] Given that AHA incorporation has been widely used in a variety of different cell types (Dieterich et al., 2010; Dieterich et al., 2006; Erdmann et al., 2015; Glenn et al., 2017; Hinz et al., 2012; Link et al., 2004; Ma and Yates, 2018), the approach described here with *Xenopus laevis* oocytes should be readily applicable to studies in other cellular expression systems.

2) Related to the above point- I would ask for a discussion of the drawbacks to the method, such as the fact that a protein of interest may contain multiple accessible cysteine and methionine residues, or that background labeling of cysteines and methionines in other proteins may make the approach intractable in cases where the protein of interest does not express in sufficient quantities similar to Kv expression in oocytes.

We have added the requested discussion (see the above response to Essential revision 1).

3) The authors extensively describe the measurement and correction of the observed FRET signals, but the purpose of this experiment was not evident. If the goal was to measure distances of donor and acceptor, it is necessary to calculate distances/distance changes, and to compare the accuracy of the chosen method as a molecular ruler to that of other approaches. As it currently stands, the significance of the FRET experiments in this study is unclear. In fact, it is not obvious if the pair of fluorophores would even be suitable for measuring distances, given the direct excitation of the acceptor. Furthermore, the authors should discuss the implications of the slow and thus likely incomplete/asymmetric labelling via the SPAAC-based method in more detail, especially because their measurements can report on more than one fluorophore pair (as shown in Figure 8—figure supplement 1).

We have revised the Results section to clarify that the rationale for the FRET experiments was to demonstrate two-color labeling of individual Shaker Kv channels. We are aware that this pair of fluorophores is not ideal for measuring distances between M356 and S424 positions and that is why we specifically do not calculate distances.

4) Comparing to the cysteine based method the AHA method shows lower efficiency of introducing fluorescent probes (Figure 5C, D). The authors pointed out this result in the text. However, it may be more helpful if more details of how this relatively low efficiency affected experimental processes can be described and troubleshooting procedures are discussed. A discussion of other precautions and possible pitfalls in using this method may be also helpful.

We have added the requested discussion (see the above response to Essential revision 1).

5) As pointed out by the authors, unnatural amino acid Anap has been used in studying ion channels, but it is known in the field that not all ion channels are suitable for the Anap approach. It will be nice to show that the AHA method can be used in some channels other than Shaker. Nevertheless, it may be out of the scope of this manuscript, and the ion channel field will find out the answer after the publication of this manuscript. We suggest that the title is modified to "Exploring structural dynamics of a membrane protein by…".

We completely agree and we are in the process of applying our approaches to other channels, but we currently do not have any results that are ready for publication and the present manuscript is already quite large. We have modified the title in the revised manuscript as suggested.

6) Combining click chemistry and cysteine labeling approaches for dual color labeling as needed for FRET has been described before (e.g. https://www.ncbi.nlm.nih.gov/pubmed/23317903). That said, I do think the paper would be of interest to the ion channel community in general, many of whom may not be familiar with these approaches. Especially the use of AHA to substitute methionine residues for later click labeling, which I suspect is less appreciated in the ion channel field. However, I would ask the authors to generally tone down their statements regarding the novelty of the approach.

We have now revised the Results and Discussion to include all the relevant references and appropriately stated the results with respect to previous work. We agree that combinations of click chemistry reactions (both SPAAC and CuAAC) and cysteine mutagenesis have been used for dual color labeling of proteins previously. However, till date, these approaches have only been applied to purified soluble proteins containing azide or alkyne terminated amino acids (CuAAC – Seo et al., 2011, Tyagi et al., 2013) (SPAAC- Sadoine et al., 2017). As far as we are aware, our study is the first to show that the SPAAC reaction can be combined with cysteine mutagenesis to carry out site-specific two-color labeling of a membrane protein expressed and studied in live cells. The CuAAC reaction involves a catalyst containing Cu^+2^ salts, reducing agent and a stabilizing ligand whereas SPAAC is a catalyst-free reaction. In the future we hope to explore whether a combination of CuAAC and cysteine mutagenesis is suitable for dual color labeling of proteins in live cells.